# Importance and Coherence: Methods for Evaluating Modularity in Neural Networks

## Abstract

As deep neural networks become more widely-used, it is important to understand their inner workings. Toward this goal, modular interpretations are appealing because they offer flexible levels of abstraction aside from standard architectural building blocks (e.g., neurons, channels, layers). In this paper, we consider the problem of assessing how functionally interpretable a given partitioning of neurons is. We propose two proxies for this: *importance* which reflects how crucial sets of neurons are to network performance, and *coherence* which reflects how consistently their neurons associate with input/output features. To measure these proxies, we develop a set of statistical methods based on techniques that have conventionally been used for the interpretation of individual neurons. We apply these methods on partitionings generated by a spectral clustering algorithm which uses a graph representation of the network's neurons and weights. We show that despite our partitioning algorithm using neither activations nor gradients, it reveals clusters with a surprising amount of importance and coherence. Together, these results support the use of modular interpretations, and graph-based partitionings in particular, for interpretability.

## 1 Introduction

Deep neural networks have achieved state-of-the-art performance in a variety of applications, but this success contrasts with the challenge of making them more intelligible. As these systems become more advanced and widely-used, there are a number of reasons we may need to understand them more effectively. One reason is to shed light on better ways to build and train them. A second reason is the importance of transparency, especially in settings which involve matters of safety, trust, or justice (Lipton, 2018). More precisely, we want methods for analyzing a trained network that can be used to construct semantic and faithful descriptions of its inner mechanisms. We refer to this as *mechanistic transparency*.

Toward this goal, we consider modularity as an organizing principle to achieve mechanistic transparency. In the natural sciences, we often try to understand things by taking them apart. Aside from subdivision into the standard architectural building blocks (e.g., neurons, channels, layers), are there other ways a trained neural network be meaningfully "taken apart"? We aim to analyze a network via a partitioning of its neurons into disjoint sets with the hope of finding that these sets are "modules" with distinct functions. Since there are many choices for how to partition a network, we would like metrics for anticipating how meaningful a given partition might be.

Inspired by the field of program analysis (Fairley, 1978), we apply the concepts of "dynamic" and "static" analysis to neural networks. Dynamic analysis includes performing forward passes and/or computing gradients, while static analysis only involves analyzing architecture and parameters. In a concurrent submission (Anonymous et al., 2021), we use spectral clustering to study the extent to which networks form clusters of neurons that are highly connected internally but not externally and find that in many cases, networks are structurally clusterable. This approach is static because the partitioning is produced according to the network's weights only, using neither activations nor gradients. Here, we build off of this concurrent submission by working to bridge graph-based clusterability and functional modularity.

To see how well neurons within each cluster share meaningful similarities, we introduce two proxies: *importance* and *coherence*. Importance refers to how crucial clusters are to the network's perfor-

mance overall and lends insight into how well a partition identifies clusters that are individually key to the network's function. Coherence refers to how consistently the neurons within a cluster correspond in their activations to particular features in data. We analyze coherence both with respect to input features and output labels. To measure these proxies, we utilize dynamic interpretability methods that have been conventionally used for single-neuron analysis to the study of these partitions. We conduct a set of experiments and hypothesis tests in networks scaling from the MNIST to the ImageNet level. In doing so, we show that spectral clustering is capable of identifying functionally important and coherent clusters of neurons. This new finding the and methods we present for combining spectral clustering with dynamic methods supports the use of modular decompositions of neurons toward mechanistic transparency.

Our key contributions are threefold:

1. Introducing two proxies, importance and coherence, to assess whether a given partitioning of a network exhibits modularity.

2. Quantifying these two proxies with interpretability methods equipped with statistical hypothesis testing procedures.

3. Applying our methods on the partitions produced by the spectral clustering technique of Anonymous et al. (2021) on a range of networks, and finding evidence of modularity among these clusters.

## 2   GENERATING PARTITIONINGS WITH SPECTRAL CLUSTERING

In our concurrent submission, we introduce and study in-depth a procedure to partition a neural network into disjoint clusters of neurons (Anonymous et al., 2021) based only on its weights. We found that trained networks are more clusterable than randomly initialized ones, and they are also often more clusterable than similar networks with identical weight distributions. The experimental procedure consists of three steps: (1) *"Graphification"* - transforming the network into an undirected edge-weighted graph; (2) *Spectral clustering* - obtaining a partitioning via spectral clustering of the graph.

**Graphification:** To perform spectral clustering, a network must be represented as an undirected graph with non-negative edges. For MLPs (multilayer perceptrons), each graph vertex corresponds to a neuron in the network including input and output neurons. If two neurons have a weight connecting them in the network, their corresponding vertices are connected by an edge giving its absolute value. For CNNs (convolutional neural networks), a vertex corresponds to a single feature map (which we also refer to as a "neuron") in a convolutional layer. Here, we do not use input, output, or fully-connected layers. If two feature maps are in adjacent convolutional layers, their corresponding vertices are connected with an edge giving the $L_1$ norm for the corresponding 2 dimensional kernel slice. If convolutional layers are separated by a batch normalization layer (Ioffe & Szegedy, 2015), we multiply weights by $\gamma/(\sigma + \varepsilon)$ where $\gamma$ is the scaling factor, $\sigma$ is the moving standard deviation, and $\varepsilon$ is a small constant.

**Spectral Clustering:** We run normalized spectral clustering on the resulting graph (Shi & Malik, 2000) to obtain a partition of the neurons into clusters. For all experiments, we set the number of clusters to 12 unless explicitly mentioned otherwise. We choose 12 because (1) it is computationally tractable, (2) it is larger than the number of classes in MNIST and CIFAR-10, and (3) it is small compared to the number of neurons in the layers of all of our networks. However, in Appendix A.6, we show results for $k = 8$ and $k = 18$ for a subset of experiments and find no major differences. We use the *scikit-learn* implementation (Pedregosa et al., 2011) with the ARPACK eigenvalue solver (Borzì & Borzì, 2006). Refer to appendix A.1 for a complete description of the algorithm.

## 3   EVALUATION OF MODULARITY USING IMPORTANCE AND COHERENCE

Clusters of neurons produced by spectral clustering span more than one layer. However, layers at different depths of a network tend to develop different representations. To control for these differences, we study the neurons in clusters separately per layer. We call these sets of neurons within the same cluster and layer "sub-clusters." In our experiments, we compare these sub-clusters to other sets of random units of the same size and same layer. When discussing these experiments, we refer

to the sub-clusters from the clustering algorithm as "true sub-clusters" and the sets composed of random neurons as "random sub-clusters." Random sub-clusters form the natural control condition to test whether the specific partitioning of neurons exhibits importance or coherence compare to alternative partitions, while taking account location and size.

As outlined in the Introduction, we study importance: how crucial each sub-cluster is to the network; input coherence: how well neurons in a sub-cluster associate with similar input features; and output coherence, how well they associate with particular output labels, as proxies for modularity. In this section, we present two types of experiments. First, we use visualization techniques on sub-clusters to measure input and output coherence, and second, we use "lesion tests" based on dropping out neurons in a sub-cluster to measure output coherence and importance.

These techniques are scalable, and we experiment with a wide range of networks. For small-scale experiments, we train and analyze MLPs with four hidden layers of 256 neurons each and small convolutional networks with 3 layers of 64 neurons each followed by a dense layer of 128 neurons trained on the MNIST (LeCun et al., 1998) and Fashion-MNIST (Xiao et al., 2017) datasets. At a mid scale, we train and analyze VGG-style CNNs containing 13 convolutional layers using the architectures from Simonyan & Zisserman (2014) trained on CIFAR-10 (Krizhevsky et al., 2009) using the procedure from Liu & Deng (2015). Finally, for the ImageNet (Krizhevsky et al., 2009) scale, we analyze pretrained ResNet18, ResNet50, (He et al., 2016) VGG-16, and VGG-19 (Simonyan & Zisserman, 2014) models.

In our concurrent submission (Anonymous et al., 2021) we show that in some cases, weight pruning and dropout can each be used to promote graph-based clusterability. We use pruning in small MLPs but no other networks. We use dropout for MLPs in correlation-based visualization experiments in subsection 3.1.1 but no other MLPs. Also, for the mid-sized VGG-CNNs, we experiment both with versions that are unregularized and which are regularized using dropout and $L_2$ regularization as done in Liu & Deng (2015). Complete training details including testing accuracies are in the appendix A.2.

## 3.1 FEATURE VISUALIZATION

### 3.1.1 CORRELATION-BASED VISUALIZATION

First, we introduce here a simple method to provide visual examples and build intuition. In later subsections, we present a quantitative approach with statistical hypothesis testing. A simple way to visualize a sub-cluster is to identify what input features each of its neurons respond to and then use these to create an aggregated visualization. We do this for small MLPs in which we construct visualizations of neurons using their correlations with the input pixels across the test dataset. We use their post-ReLU activations, and consider the activation of a convolutional feature map to be its $L_1$ norm. Instead of linear correlation, we use the Spearman correlation (which is the linear correlation of ranks) because it is able to capture relationships which tend to monotonically increase even if they are nonlinear.

After obtaining visualizations for each neuron in a sub-cluster, we do not directly take their average to visualize the entire sub-cluster. To see why, consider two neurons which are highly anticorrelated across the testing set. These neurons are highly coherent, but averaging together their visualizations would obscure this by cancellation. To fix this problem, we align the signs of the visualizations for individual neurons using a variant of an algorithm from Watanabe (2019). To visualize a sub-cluster, for a number of iterations (we use 20), we iterate over its neurons, and calculate for each the sum of cosines between its visualization and each of the other neurons' visualizations in vector form. If this sum is negative, we flip the sign of this neuron's visualization. Refer to appendix A.3 for a complete algorithmic description. After this procedure, we take the mean of the visualizations within a sub-cluster.

To see how much meaningful input coherence these sub-clusters exhibit, we compare them to random sub-clusters (recall each of these are randomly selected sets of neurons of the same size from the same layer as a true sub-cluster). Figure 1a-b shows results from MLPs trained on MNIST and Fashion-MNIST. Here, these MLPs are trained with dropout which we found to be helpful for clearer visualizations. In the first row of each image are visualizations for true sub-clusters, and the bottom four rows show visualizations for random ones. The true sub-clusters in the top row

produce more coherent visualizations with better-defined and higher-contrast features compared to the random ones in the bottom 4 rows.

Next, we hypothesized that if we trained a network on a task that lent itself well to parallel processing, spectral clustering would capture specialized modules. To test this, we designed "halves-same" and "halves-diff" tasks for small MLPs based on the MNIST and Fashion-MNIST datasets. For the halves-same tasks, two images of the same class were resized to have half their original width and concatenated side-by-side in order to create a composite image of the same size as the originals. We gave these images the same label as their component halves. For the halves-diff tasks, this was done with two images from random classes, and the resulting image was labeled with the sum of their labels modulo 10. Example images from each of the the halves-same/diff MNIST and Fashion-MNIST datasets are shown in figure 3. We expected that the halves-diff task would be more economical to compute in a modular way by separately recognizing the two halves and computing their modular sum. In appendix A.3, we show that our networks can compute this modular sum.

Figure 1c-d shows these visualizations for MLPs trained with dropout on halves-same MNIST and without dropout on halves-diff MNIST. We did not use dropout to train the halves-diff networks because it resulted in poor accuracy. This is likely because while amenable to image classification, dropout is not amenable to modulo arithmetic. Columns are arranged from left to right in the order of the layer in which they appear in the network. Visualizations for the halves-same networks tend to result in similar left and right halves, but in the early (leftmost) layers of the networks trained on the halves-diff tasks, there is a tendency for true sub-clusters to be selective to one half.

(a)

(b)

(c)

(d)

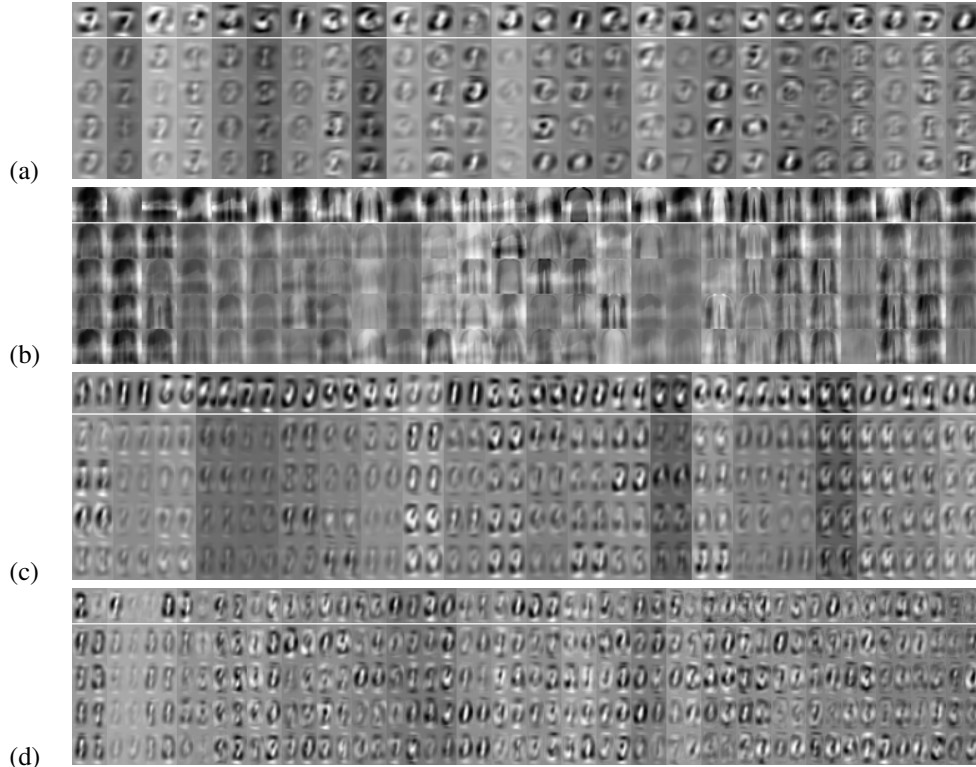

Figure 1: **Sub-cluster visualizations for small MLPs:** (a) MNIST, (b) Fashion-MNIST, (c) halves-same MNIST, (d) halves-diff MNIST. In each image, the top row gives visualizations of true sub-clusters and the bottom four for random ones of the same size. Columns are ordered left to right according to layer order. Pixel values are scaled from black to white for each column independently. All except (d) were trained with dropout.

This method of understanding input coherence has the advantage of being able to provide intuitive visual examples and efficiently construct interpretable features for MLPs. However, it was not as effective for CNNs. In appendix A.3 we detail this process, and in figure 4, we show visualizations for small CNNs in which we find less evidence of coherence among sub-clusters. To expand on

the intuitive visual examples offered here, in the following section, we introduce a more versatile, scalable method along with hypothesis testing procedures for obtaining quantitative results.

### 3.1.2 INPUT CONSTRUCTION

Another way to visualize neurons in a network is to use gradient-based optimization to create an input image which maximizes the activation of a neuron, or in our case, a sub-cluster of them. Patterns in the resulting visualizations can suggest what features the neurons respond to. We visualize sub-clusters with this method (Olah et al., 2017) using the Lucid[1] package. Implementation details are in appendix A.8. Figure 5 gives example visualizations.

To obtain quantitative results, we used two techniques. First, we analyzed the value of the maximization objective for each image we produced, which we call the "score." This gives of one notion of how coherent a sub-cluster may be with respect to input features, because if a single image can activate an entire sub-cluster well, this suggests that the neurons comprising it can be activated by similar features. Second, we analyze the entropy of the softmax outputs of the network when these images are passed through it. If the entropy of the softmax distribution is low, this suggests that a cluster is coherent with respect to outputs.

We then test the null hypothesis that these sub-clusters are equally coherent as random sets of neurons. For each sub-cluster in a network with at least three neurons and at most 80% of the neurons in a layer, we compare its visualization's score and output entropy to those of 9 random sub-clusters. We then obtain one-sided $p$ values by taking the percentiles for the true sub-cluster's score and entropy relative to the random sub-clusters' score and entropy. We take right-sided $p$ values for scores and left-sided $p$ values for output entropies so that lower $p$ values indicate greater input/output coherence in both cases. We then use two different methods to combine all sub-cluster $p$ values to obtain a combined $p$ value for the entire network for either score or entropy. Both are presented here, but full details for both are in appendix A.4.

**Fisher Method:** First, we center the sub-cluster $p$ values around 0.5 to obtain a granular approximation of the uniform distribution under the null, and then use the Fisher Method. The test statistic for a set of sub-cluster $p$ values $p_1...p_n$ is $-2\sum_{i=1}^{n} \log p_i$ which takes a chi squared distribution with $2n$ degrees of freedom under the null hypothesis.

**Chi Squared Method:** Second, since there are only a set number, $m$, of values which the $p$ values can take (in our case $m = 10$), we perform a Chi Squared categorical test to see whether their distribution among these discrete values is nonuniform. The test statistic is $\sum_{i=1}^{m} \frac{(x_i - \mu_i)^2}{\mu_i}$ in which each $x_i$ gives an observed count and each $\mu_i$ gives a expected one. It will have a chi squared distribution with $m - 1$ degrees of freedom under the null hypothesis.

These methods test for different things. The Fisher method indicates how low the $p$ values for sub-clusters tend to be across a network and tests whether the true sub-clusters are consistently more coherent than random ones. However, the distribution of sub-cluster $p$ values may be nonuniform but in a way that the Fisher Method is not designed to detect. For example, they may tend to be very high or follow a U-shaped distribution. The Chi Squared method adds additional resolution by detecting cases like this.

The top section of table 1 summarizes these results. For each network which we perform this test on, we provide the Fisher and Chi Squared categorical $p$ values for both the score (input coherence) and output entropy (output coherence). For the non-ImageNet networks, we report results for each measure (separately) as a median across 5 networks. We find strong evidence of significant levels of input coherence in the VGG family of networks, and find that the unregularized VGGs trained on CIFAR-10 also seems to exhibit a significant amount of output coherence. In Appendix A.8, we also present experiments for understanding variance of activations in true and random sub-clusters.

### 3.2 LESION TESTS

Another set of tools that has been used for understanding both biological (Gazzaniga & Ivry, 2013) and artificial (Zhou et al., 2018; Casper et al., 2020) neural systems involves disrupting neurons dur-

---

[1]https://github.com/tensorflow/lucid

| | Visualization Score: Input Coherence | | Softmax Entropy: Output Coherence | |
|---|---|---|---|---|
| **Network** | **Fisher Combined** $p$ | **Chi Squared** $p$ | **Fisher Combined** $p$ | **Chi Squared** $p$ |
| MLP MNIST | 0.575 | 0.810 | 0.377 | 0.365 |
| CNN MNIST | 0.671 | 0.328 | 0.504 | 0.135 |
| VGG CIFAR (Unreg) | $\mathbf{3.45 \times 10^{-4}}$ | $\mathbf{8.79 \times 10^{-9}}$ | $\mathbf{2.00 \times 10^{-7}}$ | $\mathbf{1.29 \times 10^{-23}}$ |
| VGG CIFAR | **0.035** | $\mathbf{3.19 \times 10^{-5}}$ | 0.891 | 0.082 |
| VGG-16 ImageNet | $\mathbf{2.54 \times 10^{-4}}$ | $\mathbf{1.41 \times 10^{-7}}$ | 0.118 | 0.281 |
| VGG-19 ImageNet | **0.003** | $\mathbf{5.69 \times 10^{-4}}$ | 0.633 | 0.888 |
| ResNet50 ImageNet | 0.355 | 0.437 | 0.435 | 0.812 |

| | Accuracy Change: Importance | | Classwise Range: Output Coherence | |
|---|---|---|---|---|
| **Network** | **Fisher Combined** $p$ | **Chi Squared** $p$ | **Fisher Combined** $p$ | **Chi Squared** $p$ |
| MLP MNIST | $\mathbf{1.34 \times 10^{-8}}$ | $\mathbf{4.40 \times 10^{-14}}$ | 0.968 | **0.035** |
| MLP Fashion | $\mathbf{7.73 \times 10^{-6}}$ | $\mathbf{1.76 \times 10^{-11}}$ | 0.513 | **0.003** |
| CNN MNIST | **0.013** | **0.035** | 0.480 | 0.561 |
| CNN Fashion | **0.025** | 0.483 | 0.738 | 0.283 |
| VGG CIFAR (Unreg) | $\mathbf{2.25 \times 10^{-11}}$ | $\mathbf{4.30 \times 10^{-44}}$ | $\mathbf{3.19 \times 10^{-10}}$ | $\mathbf{3.47 \times 10^{-33}}$ |
| VGG CIFAR | $\mathbf{7.17 \times 10^{-6}}$ | $\mathbf{1.55 \times 10^{-17}}$ | $\mathbf{5.49 \times 10^{-6}}$ | $\mathbf{7.93 \times 10^{-26}}$ |
| ResNet-18 ImageNet | **0.002** | $\mathbf{1.51 \times 10^{-5}}$ | 0.112 | **0.001** |
| VGG-16 ImageNet | $\mathbf{9.81 \times 10^{-13}}$ | $\mathbf{1.56 \times 10^{-40}}$ | 1.000 | $\mathbf{1.62 \times 10^{-23}}$ |

Table 1: **Combined** $p$ **values for feature visualization and lesion experiments:** Each $p$ value is a combination of $p$ values for all of a network's sub-clusters created with the Fisher or Chi Squared method. For ImageNet networks, the values reflect results for single networks, but for all others, values (separately) give a median across 5 independently trained replicates. We do not ascribe any particular epistemic significance to the threshold of 0.05 (also note that significance thresholds normally need to be adjusted when taking a median of $p$ values) but we bold all $p$ values less than it. In Appendix A.7, we present corrections for multiple comparisons. In Appendix A.6, we show results for $k = 8$ and $k = 18$. **Feature Visualization (Top):** Results indicate tests for visualizations produced with Lucid. **Lesions (Bottom):** Results indicate tests involving lesions of sub-clusters. Due to computational demands, we limit our analysis here at the ImageNet scale to ResNet-18 and VGG-16 models.

ing inference. Whereas the images produced with feature visualization were optimized to maximally activate a sub-cluster, we perform a dual type of experiment with "lesion" tests in which we analyze network outputs when a sub-cluster is dropped out. When lesioning a sub-cluster, we set all weights incoming to the constituent neurons to 0, while leaving the rest of the network untouched. Refer to figure 6 for example plots of the accuracy drops for a small MLP and CNN trained on Fashion-MNIST. We then determine the damage to the network's overall and per-class testing accuracy. This allows us to evaluate both importance and output coherence.

**Importance** Importance allows us to identify which sub-clusters are key to the network and therefore of particular interest for scrutiny. To systematically quantify the importance of a network's sub-clusters in aggregated way, we combine the right-sided $p$ values of all the network's true sub-clusters using the Fisher and Chi Squared categorical methods discussed above 3.1.2 and described in detail in appendix A.4. Note that these experiments are analogous to those in 3.1.2 The bottom section of table 1 gives results for these. We find strong evidence across the networks which we train that spectral clustering reveals important sub-clusters.

There is generally significant diversity among sub-clusters in their size, importance, and importance relative to random sub-clusters. To demonstrate this, we construct an example *descriptive* taxonomy, in which we consider three criteria for identifying particularly important sub-clusters. First, the sub-cluster should be at least 5% of the neurons of the layer. Second, the drop in accuracy under lesion should be greater than 1 percentage point; and third, the drop should not simply be due to the number of damaged neurons. To evaluate the third criterion, we generate random sub-clusters with the same number of neurons as the true sub-cluster from the same layer, and collect the distribution of accuracy drops. We say that this criterion is met if the accuracy drop for the true sub-cluster is greater than all of 20 random sub-clusters, i.e its $p$ value is smaller than 1/20.

In figure 2, we plot sub-cluster size versus accuracy drop for an MLP trained on Fashion-MNIST and a VGG trained on CIFAR-10 that has been clustered into 8 clusters (we use 8 for the sake of visualization here, but we use 12 clusters for all quantitative experiments). Many sub-clusters are too small to be counted as important, and many are significantly impactful compared to random sub-clusters but not practically significant. However, some clearly are practically important for the functioning of the network.

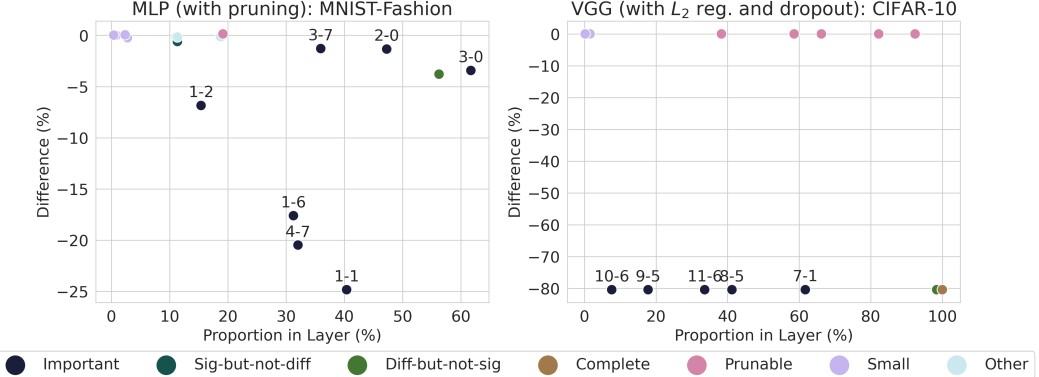

Figure 2: **Plot of different sub-clusters of two networks.** Important sub-clusters are labeled first by their layer number and then their cluster number. The horizontal axis shows the proportion of their layer's neurons in the sub-cluster, and the vertical axis shows the reduction in accuracy from lesioning it. 'Important' means that the sub-cluster makes up at least 5% of the layer, that the drop in accuracy is greater than one percentage point, and that it was more important than all of 20 random sub-clusters it was compared against—that is to say, statistically significant. 'Sig-but-not-diff' means that the drop in accuracy is statistically significant but less than 1 percentage point, 'Diff-but-not-sig' means that the lesioning damage was more than 1 percentage point but not statistically significant, 'Prunable' means that the drop in the accuracy is smaller than all random shuffles and smaller than 1 percentage point, 'Complete' means that the sub-cluster contains the whole layer, 'Small' means that the sub-cluster consists of less than 5% of the layer, and 'Other' means that the drop in accuracy is not statistically significant and less than 1 percentage point. Both of the networks are partitioned into 8 clusters. The data is based on two tables which are included in appendix A.9. Note that the 'Complete' category appears only in the right plot, and only a single point is shown although there are 8 such sub-clusters. Refer to appendix A.9 for additional details.)

**Coherence** To measure the output coherence using lesions, we analyze the accuracy changes for each of the output classes. For ten classes, we define $d = (d_0, d_1, \ldots, d_9)$, where $d_i$ is the change in the $i$-th class accuracy due to the lesioning of a sub-cluster. In order to obtain a measurement independent of the overall importance, we divide these class-wise accuracy changes by their mean, $d' = d/\bar{d}$, and the then take their range $\Delta = \max d' - \min d'$. We refer to this as the (normalized) class-wise range. We compare true and random sub-clusters to obtain a right-sided $p$ value for each sub-cluster based on the $p$ values of the true $\Delta$. We then combine these for the entire network using the Fisher and Chi Squared categorical methods as discussed above and detailed in appendix A.4.

These results are in the bottom section of table 1. The Chi Squared $p$ values demonstrate that spectral clustering usually identifies sub-clusters with a significantly different distribution of importances compared to random sub-clusters. Meanwhile, the Fisher tests suggests that at least in VGG networks trained on CIFAR-10, the sub-clusters exhibit more output coherence. Interestingly, for VGG-16s trained on ImageNet, the opposite seems to be the case. The Fisher $p$ value is high, suggesting that the $p$ values for its individual sub-clusters tend to be high. However, the Chi Squared $p$ value is low, suggesting nonuniformity among the sub-cluster $p$ values. Together, these indicate the the clusters are consistently less coherent than random ones.

## 4 RELATED WORK

The most closely-related work to this is our paper under concurrent submission (Anonymous et al., 2021) which uses the same spectral clustering-based approach to establish that deep networks are in many cases clusterable and investigates in depth methods can be used to control the development of clusterability. Both of these works inherit insights from network science involving clustering in general (Girvan & Newman, 2002; Newman & Girvan, 2004), and spectral clustering (Shi & Malik, 2000; von Luxburg, 2007) in particular.

Our experiments in which we combine spectral clustering with correlation-based visualization (Watanabe, 2019), feature visualization (Olah et al., 2017), and lesions (Zhou et al., 2018) highlight the usefulness of combining multiple interpretability methods in order to build an improved set of tools for more rigorously understanding systems. In a similar way, other dynamic techniques for interpretability such as analysis of selectivity (Madan et al., 2020), network "dissection" (Bau et al., 2017; Mu & Andreas, 2020), earth-mover distance (Testolin et al., 2020), or intersection information (Panzeri et al., 2017) could also be combined with static graph-based partitionings under a similar framework. There already exist examples of interpretability methods being used for the identification of unexpected adversarial weaknesses (Carter et al., 2019; Mu & Andreas, 2020). We expect that developing more powerful tools like these for scrutinizing networks will be helpful toward building more robust systems.

This work adds to a growing body of research focused on modularity and compositionality in neural systems (e.g. Lake et al. (2015; 2017); Csordás et al. (2020); You et al. (2020)). This paradigm is useful both for interpretability and for building better models. Neural circuits with distributed, non-modular representations pose a litany of challenges including non-interpretability, less useful representations, poorer generalization, catastrophic forgetting, and biological implausibility. One limitation of this work is a focus on clustering in models which have fairly monolithic architectures (e.g. all neurons/filters in one layer being connected to all neurons/filters in the next). However, there exists a body of research focused specifically on developing more modular networks which either have an explicitly-modular architecture (Alet et al., 2018; Parascandolo et al., 2018; Goyal et al., 2019) or are trained in a way that promotes modularity via regularization or parameter isolation (Kirsch et al., 2018; De Lange et al., 2019).

## 5 DISCUSSION

In this work, we introduce an approach for evaluating whether a partitioning of a network exhibits modular characteristics. Key to this is analyzing proxies: importance as a means of understanding what parts of a network are crucial for performance, input/output coherence as measures for how specialized these parts are. We measure these proxies using statistical hypothesis testing procedures based on interpretability techniques which have conventionally been used for analyzing individual neurons. Though we analyze partitions produced by spectral clustering, a static method, we find that these clusters exhibit a significant amount of importance compared to random clusters. We also show that our networks in the VGG family also tend to exhibit a significant level of input coherence, and in some cases, output coherence. By and large, these findings, and those of a concurrent submission (Anonymous et al., 2021), support the analysis of modules, and in particular graph-based clusters of neurons, for developing a better understanding of neural networks' inner-workings.

Building a framework for evaluating modularity in neural networks can can guide the development of new interpretability methods which examine networks at the module level. Toward this goal, compositionality, how modules are combined and interact together, can be another proxy of modularity. For evaluating this, some of our methods can be extended to study dependencies between clusters. In appendix A.10, we present exploratory lesion-based experiments for studying cluster interactions and constructing dependency graphs.

While we make progress here toward mechanistic transparency, neural systems are still complex, and more insights are needed to develop richer understandings. The ultimate goal would be to master the process of building compositional systems which lend themselves to simple and faithful semantic interpretations. We hope that using modularity as an organizing principle to achieve mechanistic transparency and expanding our interpretability toolbox with combined static and dynamic methods will lead to a richer understanding of networks and better tools for building them to be reliable.

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

# A APPENDIX

## A.1 SPECTRAL CLUSTERING ALGORITHM

The spectral clustering algorithm on the graph $G = (V, E)$ produces a partition of its vertices, in which there are stronger connections within sets of vertices than between them (Shi & Malik, 2000). It does so by approximately minimizing the *n-cut* (normalized cut) of a partition. For disjoint, non-empty sets $X_1, ...X_k$ where $\cup_{i=1}^{k} X_i = V$, it is defined by (von Luxburg, 2007) as:

$$\text{n-cut}(X_1, ..., X_k) := \frac{1}{2} \sum_{i=1}^{k} \frac{W(X_i, \overline{X_i})}{\text{vol}(X_i)}$$

where $A := (w_{ij})_{i,j=1..n}$ is the adjacency matrix of the graph $G$; for two sets of vertices $X, Y \subseteq V$, we define $W(X, Y) := \sum_{v_i \in X, v_j \in Y} w_{ij}$; the degree of a vertex $v_i \in V$ is $d_i = \sum_{j=1}^{n} w_{ij}$; and the volume of a subset $X \subseteq V$ is $\text{vol}(X) := \sum_{i \in X} d_i$.

---

**Algorithm 1: Normalized spectral clustering** according to (Shi & Malik, 2000), implemented in *scikit-learn* (Pedregosa et al., 2011), description taken from von Luxburg (2007).

**Input** : Weighted adjacency matrix $W \in \mathbb{R}^{n \times n}$, number $k$ of clusters to construct
1 Compute the unnormalized Laplacian $L$.
2 Compute the first $k$ generalized eigenvectors $u_1, ..., u_k$ of the generalized eigenproblem $Lu = \lambda Du$.
3 Let $U \in \mathbb{R}^{n \times k}$ be the matrix containing the vectors $u_1, ..., u_k$ as columns.
4 For $i = 1, .., n$, let $y_i \in \mathbb{R}^k$ be the vector corresponding to the $i^{\text{th}}$ row of $U$.
5 Cluster the points $(y_i)_{i=1,...,n}$ in $\mathbb{R}^k$ with the *k-means* algorithm into clusters $C_1, ..., C_k$,
**Output:** Clusters $A_1, ..., A_k$ with $A_i = \{j | y_j \in C_i\}$.

---

## A.2 NETWORK TRAINING DETAILS

We use Tensorflow's implementation of the Keras API Abadi et al. (2015); Chollet et al. (2015). When training all networks, we use the Adam algorithm (Kingma & Ba, 2014) with the standard Keras hyperparameters: learning rate 0.001, $\beta_1 = 0.9$, $\beta_2 = 0.999$, no amsgrad. The loss function was categorical cross-entropy.

**Small MLPs (MNIST and Fashion-MNIST):** We train MLPs with 4 hidden layers, each of width 256, for 20 epochs of Adam (Kingma & Ba, 2014) with batch size 128. We then prune on a polynomial decay schedule (Zhu & Gupta, 2017) up to 90% weight-sparsity for an additional 20 epochs after initial training. Initial and final sparsities were chosen due to their use in the TensorFlow Model Optimization Tutorial.[2] In cases where we use dropout (for correlation visualization experiments including halves-diff tasks), we apply it after each fully-connected layer with a rate of 0.5. All MLPs achieved a testing accuracy on the MNIST and Fashion-MNIST datasets of at least 97% and 86% respectively except for the ones trained on the Halves-diff datasets which all achieved an accuracy of at least 92% and 71% respectively.

**Small CNNs (MNIST and Fashion-MNIST):** These networks had 3 convolutional layers with 64 $3 \times 3$ channels each with the second and third hidden layers being followed by max pooling with a 2 by 2 window. There was a final fully-connected hidden layer with 128 neurons. We train them with a batch size of 64 for 10 epochs with no dropout or pruning. All small CNNs achieved a testing accuracy on the MNIST and Fashion-MNIST datasets of at least 99% and 89% respectively except for the ones trained on the Halves-diff datasets which all achieved an accuracy of at least 89% and 67% respectively.

**Mid-sized VGG CNNs (CIFAR-10):** We implement a version of VGG-16 described by Simonyan & Zisserman (2014); Liu & Deng (2015). We train these with Adam, and $L_2$ regularization with

---

[2]URL: `https://web.archive.org/web/20190817115045/https://www.tensorflow.org/model_optimization/guide/pruning/pruning_with_keras`

a coefficient of $5 \times 10^{-5}$ for 200 epochs with a batch size of 128. Training was done with data augmentation which consisted of random rotations between 0 and 15 degrees, random shifts both vertically and horizontally of up to 10% of the side length, and random horizontal flipping. In cases where we use dropout, we use a per-layer dropout rate as specified in Liu & Deng (2015). All of these networks achieved testing accuracies of at least 87%.

**Large CNNs (ImageNet):** We experimented with VGG-16 and 19 (Simonyan & Zisserman, 2014) and ResNet-18, and 50 (He et al., 2016) networks. Weights were obtained from the Python `image-classifiers` package, version 1.0.0.

### A.3 CORRELATION-BASED VISUALIZATION

---

**Algorithm 2:** Sign Alignment Algorithm (Similar to Watanabe (2019))

---

**Result:** Set of sign-aligned neuron visualizations.
**Input** Neuron visualizations $V_{1:n}$ **for** *iter in num_iters* **do**
    **for** $v_i$ *in* $V$ **do**
        Calculate sum of cosines, $c = \sum_{j \neq i} \frac{v_i \cdot v_j}{\sqrt{v_i \cdot v_i} \sqrt{v_j \cdot v_j}}$
        **if** $c < 0$ **then**
            $v_i \leftarrow -v_i$
        **end**
    **end**
**end**

---

Algorithm 2 gives the sign alignment algorithm we use which is based on a a similar one from Watanabe (2019).

Figure 3 shows examples from the 'halves' and 'stack' datasets which we use for MLPs and CNNs respectively to test whether a parallelizable task can cause a network to develop clusters that cohere with one portion of the inputs or another. Details of the halves dataset experiments are detailed in Section 3. Analogous experiments for CNNs were done but with "stack" datasets. For CNNs with max pooling, object detection is insensitive to spatial location, so we design stack-same and stack-diff datasets in an analogous way using channels instead of image-halves.

Visualizations for sub-clusters in the halves datasets are provided in section 3. However, here in figure 4 are visualization results for Small CNNs for the stack-same/diff datasets. Unlike for the small MLPs, these visualizations do show obvious coherence among clusters, which was part of our motivation for the subsequent input construction experiments.

For constructing all correlation-based visualizations, we use the Spearman correlation which is defined as the linear (Pearson) correlation of ranks. This measures how one series of values of values can be expressed as a monotonically increasing function of another. We used this rather than linear correlation because of the nonlinear nature of deep networks.

**Networks can Compute Modular Sums:** A network an do this for $M$ values by using an intermediate layer of $M^2$ neurons, each of which serve as a detector of one of the possible combinations of inputs. Consider a ReLU MLP with $2M$ inputs, a single hidden layer with $M^2$ neurons, and then $M$ outputs. Suppose that it is given the task of mapping datapoints in which the input nodes numbered $i$ and $M + j$ are activated with value 1 to an output in which the $\mod (i + j, M)^{\text{th}}$ node is active with value 1. It could do so if each hidden neuron with a ReLU activation detected one of the $M^2$ possible input combinations via a bias of -1 and two weights of 1 connecting it to each of the input nodes in the combination is detects. A single weight from each hidden neuron to its corresponding output point would allow the network to compute the modular sum. In our networks, we have $M = 10$ classes, and all MLPs and CNNs have a dense layer with $> 10^2$ neurons preceding the output layer. Thus, they are capable of computing a modular sum in the halves and stack-diff tasks we give to them.

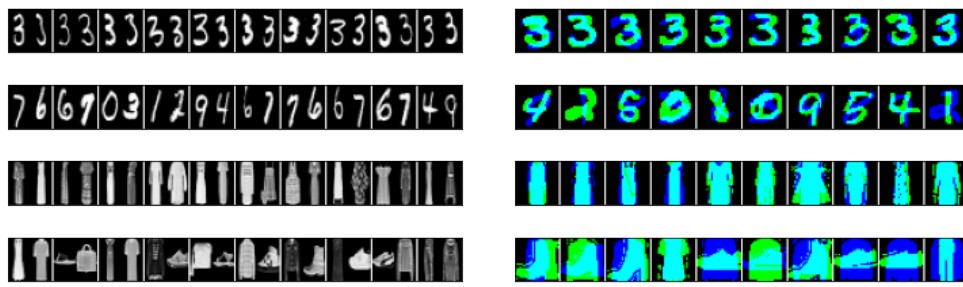

Figure 3: **Examples images from halves/stack datasets:** (Left) Samples from 'halves' datasets, and (Right) samples from 'stack' datasets, all of class 3. Each row has 10 images from the respective dataset. The first row is MNIST halves/stack-same, second is MNIST halves/stack-diff, third is Fashion halves/stack-same, and fourth is Fashion halves/stack-diff.

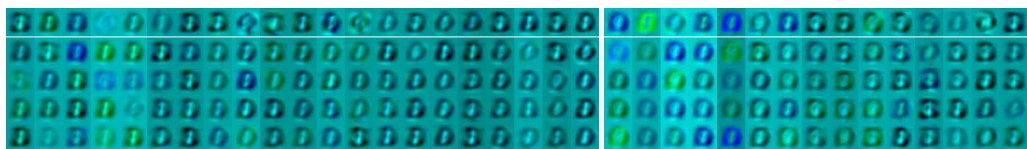

Figure 4: **Sub-cluster visualizations for small CNNs trained on stack-same/diff MNIST:** (Left) stack-same MNIST, (Right) stack-diff MNIST. The top rows give visualizations for true sub-clusters and the bottom four for random ones. Columns are ordered left to right according to layer order.

### A.4 HYPOTHESIS TESTING

Here, we provide details for the hypothesis testing methods used for input construction and lesion experiments in Section 3. In each of these experiments, for all sub-clusters in a network, we obtain quantities for the true sub-clusters and random sub-clusters. We compare these values to get a $p$ value in the form of a percentile for each true-cluster comparing it to the random ones. We then obtain a single combined $p$ value for a network overall using two methods. Both methods involve constructing a test statistic which has a chi squared distribution under the null hypothesis that true-sub-clusters have the same properties as random ones.

**Fisher Method:** This measures how low $p$ values for the true sub-clusters are overall across a network. In our case, because we use 9 random sub-clusters, the $p$ values for sub-clusters take values in $\{0.1, 0.2...1.0\}$. To obtain a granular approximation of the uniform distribution under the null, we subtract 0.05 from them to center their distribution around 0.5 so that they give a granular approximation to the continuous $\text{Uniform}(0, 1)$ distribution under the null hypothesis that visualizations for true sub-clusters are as good as random. Then we obtain the Fisher method test statistic

$$-2 \sum_{i=1}^{n} \log p_i$$

Which for $n$ and $p$ values has a chi squared distribution on $2n$ degrees of freedom under the null. We then conduct a right-sided test with respect to this distribution. The fact that we use a granular approximation of the uniform distribution makes this test conservative because when $-2$ times the sum of the logs of the $p$ values is taken during the calculation of the test-statistic, the smallest $p$ values will pull the test statistic toward the heavy tail of the Chi Squared distribution while the largest ones will pull it toward zero.

**Chi Squared Categorical Method:** This measures how nonuniform the distribution of $p$ values were for sub-clusters were across a network. The $p$ values fall into discrete bins, so a standard Chi Squared categorical test can be used to test to see whether the assortment across the sub-clusters for a network is consistent with randomness or not. The test statistic is

$$\sum_{i=1}^{m} \frac{(x_i - \mu_i)^2}{\mu_i}.$$

Here, this is a sum over $m$ discrete values which results can take, and each $x_i$ gives the count of observations of each value while $\mu_i$ gives the expected count under the null. This test statistic will take a Chi Squared distribution on $m-1$ degrees of freedom under the null hypothesis. We conduct a right-sided test with respect to this distribution.

## A.5 COHERENCE IN UNTRAINED NETWORKS

In most cases, we find that trained networks, exhibit significant levels of importance and/or coherence. However, in order to get a sense of how much importance and coherence result from the training process, it is also natural to ask to what extent untrained, randomly-initialized networks exhibit these. Here, we present the results for experiments with feature visualization as done in table 1a. We do not do this for lesion tests though because in expectation, any untrained network will have accuracy at the random guess baseline whether intact or lesioned. Table 2 shows these results for untrained CIFAR-10 scale VGGs. Here, the $p$ values for input coherence are not indicative of any sort of interesting phenomenon which contrasts with the corresponding input coherence values from table 1 which are very low. These suggest that the training process promotes input coherence in these networks. For output coherence, the $p$ values here are lower than the regularized VGGs but higher than the unregularized VGGs from table 1.

| | Visualization Score: Input Coherence | | Softmax Entropy: Output Coherence | |
|---|---|---|---|---|
| **Network** | **Fisher Combined** $p$ | **Chi Squared** $p$ | **Fisher Combined** $p$ | **Chi Squared** $p$ |
| VGG (Untrained) | 0.519 | 0.358 | **0.002** | **0.006** |

Table 2: **Results of Lesion tests for randomly initialized mid-sized VGG networks:** As in Table 1, each cell gives a median for 5 independent networks, and values less than 0.05 are bolded.

## A.6 LESION TESTS WITH ALTERNATE CHOICES OF $k$

In all quantitative experiments in the main paper, we present results for $k=12$ clusters. However, to test the robustness of result to the choice of $k$, we present here in table 3, replicates of table 1b with 8 (50% more) and 18 (50% fewer) clusters. Overall, results are very similar with no apparent systematic differences. In table 3a, there are only 3 values which are different from table 1b by whether they are below the threshold of 0.05, and similarly, there is only 1 such value in 3b.

| $k=8$ | **Accuracy Change: Importance** | | **Classwise Range: Output Coherence** | |
|---|---|---|---|---|
| **Network** | **Fisher Combined** $p$ | **Chi Squared** $p$ | **Fisher Combined** $p$ | **Chi Squared** $p$ |
| MLP MNIST | $\mathbf{1.59 \times 10^{-6}}$ | $\mathbf{1.05 \times 10^{-13}}$ | 0.304 | 0.098 |
| MLP Fashion | $\mathbf{1.88 \times 10^{-5}}$ | $\mathbf{8.85 \times 10^{-10}}$ | 0.586 | **0.002** |
| CNN MNIST | **0.008** | **0.028** | 0.570 | 0.587 |
| CNN Fashion | **0.005** | **0.005** | 0.700 | 0.582 |
| VGG CIFAR (Unreg) | $\mathbf{1.24 \times 10^{-8}}$ | $\mathbf{1.59 \times 10^{-35}}$ | $\mathbf{2.17 \times 10^{-9}}$ | $\mathbf{6.07 \times 10^{-32}}$ |
| VGG CIFAR | $\mathbf{3.99 \times 10^{-7}}$ | $\mathbf{7.08 \times 10^{-29}}$ | $\mathbf{7.97 \times 10^{-9}}$ | $\mathbf{1.80 \times 10^{-38}}$ |
| ResNet-18 ImageNet | 0.130 | **0.001** | 0.086 | **0.021** |
| VGG-16 ImageNet | $\mathbf{3.36 \times 10^{-5}}$ | $\mathbf{6.08 \times 10^{-8}}$ | 0.998 | **0.006** |

| $k=18$ | **Accuracy Change: Importance** | | **Classwise Range: Output Coherence** | |
|---|---|---|---|---|
| **Network** | **Fisher Combined** $p$ | **Chi Squared** $p$ | **Fisher Combined** $p$ | **Chi Squared** $p$ |
| MLP MNIST | $\mathbf{1.23 \times 10^{-7}}$ | $\mathbf{1.21 \times 10^{-16}}$ | 0.370 | **0.027** |
| MLP Fashion | $\mathbf{5.67 \times 10^{-5}}$ | $\mathbf{6.31 \times 10^{-11}}$ | 0.115 | **0.019** |
| CNN MNIST | **0.042** | 0.088 | 0.503 | 0.569 |
| CNN Fashion | **0.035** | 0.328 | 0.372 | 0.866 |
| VGG CIFAR (Unreg) | $\mathbf{2.58 \times 10^{-13}}$ | $\mathbf{1.42 \times 10^{-40}}$ | $\mathbf{1.84 \times 10^{-15}}$ | $\mathbf{1.49 \times 10^{-54}}$ |
| VGG CIFAR | $\mathbf{4.07 \times 10^{-5}}$ | $\mathbf{1.63 \times 10^{-17}}$ | $\mathbf{4.69 \times 10^{-8}}$ | $\mathbf{2.72 \times 10^{-33}}$ |
| ResNet-18 ImageNet | $\mathbf{4.08 \times 10^{-4}}$ | $\mathbf{7.50 \times 10^{-8}}$ | 0.411 | **0.001** |
| VGG-16 ImageNet | $\mathbf{2.94 \times 10^{-16}}$ | $\mathbf{1.41 \times 10^{-65}}$ | 1.000 | $\mathbf{3.34 \times 10^{-57}}$ |

Table 3: **Combined $p$ values for lesion experiments with $k=8$ (top) and $k=18$ (bottom):** Table 1b replicated with alternate choices of the number of clustering centers for the same networks.

| Network | Visualization Score: Input Coherence | | Softmax Entropy: Output Coherence | |
|---|---|---|---|---|
| | Fisher Combined $p$ | Chi Squared $p$ | Fisher Combined $p$ | Chi Squared $p$ |
| MLP MNIST | | | | |
| CNN MNIST | | | | |
| VGG CIFAR (Unreg) | √ | √ | √ | √ |
| VGG CIFAR | | √ | | √ |
| VGG-16 ImageNet | √ | √ | | |
| VGG-19 ImageNet | √ | √ | | |
| ResNet50 ImageNet | | | | |

| Network | Accuracy Change: Importance | | Classwise Range: Output Coherence | |
|---|---|---|---|---|
| | Fisher Combined $p$ | Chi Squared $p$ | Fisher Combined $p$ | Chi Squared $p$ |
| MLP MNIST | √ | √ | | |
| MLP Fashion | √ | √ | | √ |
| CNN MNIST | √ | | | |
| CNN Fashion | | | | |
| VGG CIFAR (Unreg) | √ | √ | √ | √ |
| VGG CIFAR | √ | √ | √ | √ |
| ResNet-18 ImageNet | √ | √ | | √ |
| VGG-16 ImageNet | √ | √ | | √ |

Table 4: Significance of results in table 1, using the Benjamini-Hochberg procedure to ensure that the false discovery rate per group—that is, the expectation under the data-generating distribution of the proportion of results declared significant that came from the null distribution—is below 0.05, where all the Fisher tests are grouped together and all the chi squared tests are separately grouped together.

### A.7 MULTIPLE COMPARISON ADJUSTMENT

In table 1, we report various $p$ values that summarize the degree to which statistics of sub-clusters vary from those of random groups of neurons within a network. For each network, one can use the $p$ value to test whether the sub-cluster statistics are drawn from the same distribution of the statistics of random groups of neurons. However, when testing multiple networks, one might want to ensure that the experiment and significance-testing procedure are unlikely to generate false positives. In order to do this, a more complicated procedure to decide significance must be used.

The Benjamini-Hochberg procedure (Benjamini & Hochberg, 1995) controls the false discovery rate: that is, the expected proportion of rejections of the null hypothesis that are false positives, where the expectation is taken under the data-generating distribution. It relies on all experiments being independent, and therefore it was run separately on the Fisher combined $p$ values and on the Chi squared $p$ values. Results that are declared significant under this procedure when the maximum acceptable false discovery rate is 1/20 is shown in table 4.

The Holm-Bonferroni method (Holm, 1979) controls the family-wise error rate: that is, the probability under the data-generating distribution that any null hypotheses are falsely rejected. Results that are declared significant by the Holm-Bonferroni method run on the whole of table 1 capping the family-wise error rate at 1/20 are shown in table 5.

### A.8 INPUT CONSTRUCTION

All visualizations were created using the Lucid[3] package. The optimization objective for visualizing sub-clusters was the mean post-ReLU activation for all neurons inside the cluster (it was a mean of means for convolutional feature maps). For small MLPs, small CNNs, and mid-sized CNNs, we generated images using random jittering and scaling, and for ImageNet models, we used Lucid's default transformations which consist of padding, jittering, rotation, and scaling with default hyperparameters. For all networks, we used the standard pixel-based parameterization of the image and no regularization on the Adam optimizer. For visualizations in small MLPs and CNNs, we used versions of these networks trained on 3-channel versions of their datasets in which the same

---

[3]https://github.com/tensorflow/lucid

|  | Visualization Score: Input Coherence | | Softmax Entropy: Output Coherence | |
|---|---|---|---|---|
| Network | Fisher Combined $p$ | Chi Squared $p$ | Fisher Combined $p$ | Chi Squared $p$ |
| MLP MNIST |  |  |  |  |
| CNN MNIST |  |  |  |  |
| VGG CIFAR (Unreg) | ✓ | ✓ | ✓ | ✓ |
| VGG CIFAR |  | ✓ |  | ✓ |
| VGG-16 ImageNet | ✓ | ✓ |  |  |
| VGG-19 ImageNet |  | ✓ |  |  |
| ResNet50 ImageNet |  |  |  |  |

|  | Accuracy Change: Importance | | Classwise Range: Output Coherence | |
|---|---|---|---|---|
| Network | Fisher Combined $p$ | Chi Squared $p$ | Fisher Combined $p$ | Chi Squared $p$ |
| MLP MNIST | ✓ | ✓ |  |  |
| MLP Fashion | ✓ | ✓ |  |  |
| CNN MNIST |  |  |  |  |
| CNN Fashion |  |  |  |  |
| VGG CIFAR (Unreg) | ✓ | ✓ | ✓ | ✓ |
| VGG CIFAR | ✓ | ✓ | ✓ | ✓ |
| ResNet-18 ImageNet |  | ✓ |  | ✓ |
| VGG-16 ImageNet | ✓ | ✓ |  | ✓ |

Table 5: Significance of results in table 1, using the Holm-Bonferroni method to ensure that the family-wise error rate—that is, the probability under the data-generating distribution that any result is falsely declared significant— is less than 1/20.

inputs were stacked thrice because Lucid requires networks to have 3-channel inputs. However, we show grayscaled versions of these in figure 5. Refer to the main text (section 3.1.2) for quantitative analysis of the optimization objective values.

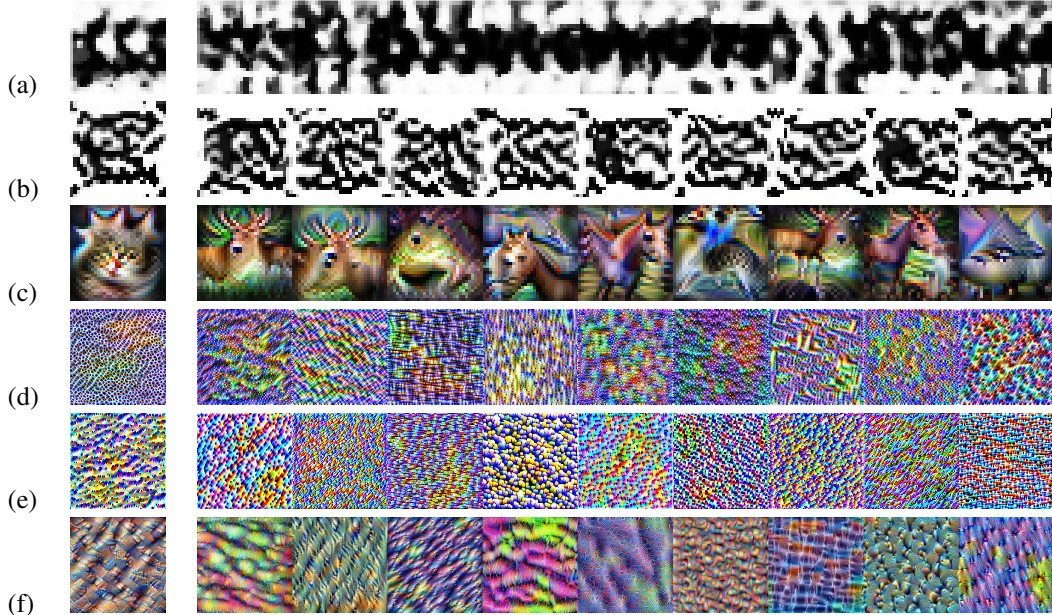

Figure 5: **Example feature visualizations for true and random sub-clusters:** In the left column are shown true sub-cluster visualizations, and in the right column are visualizations of sub-clusters of random neurons of the same size in the same layer. (a) MLP, MNIST; (b) CNN, MNIST; (c) CNN-VGG, CIFAR-10; (d) VGG-16, ImageNet; (e) VGG-19, ImageNet; (f) ResNet-50, ImageNet.

Importantly these feature visualizations, while designed to maximally activate a sub-cluster, will not necessarily highly activate all of the neurons inside of it. In order to get a sense of how much

variance there is among these activations, we analyze two properties of the distribution of sub-cluster activations when a visualization is passed through the network.

First, we perform the same tests as for score and entropy in table 1, but with the variance among neuron activations. The "Activational Variance" columns in table 6 show these $p$ values. Here, low Fisher $p$ values reflect a low variance for unit activations in a true sub-cluster compared to the variance for unit activations in random sub-clusters. In table 6, there is significant evidence that some of the networks at the CIFAR-10 and ImageNet scale have lower variance among the activations of true sub-clusters than random ones when a sub-cluster's visualization is passed through the network. This suggests that in the networks for which this is the case, neurons in true sub-clusters are more consistently activated by the same visualizations than those in random sub-clusters.

Second, we directly analyze the empirical coefficients of variation (CoVs) for the distributions of true sub-clusters. The CoV is the standard deviation of a distribution divided by its mean: $\hat{\sigma}/\hat{\mu}$. As such, a high CoV means that the distribution has a high standard deviation relative to the mean. For each sub-cluster of a network, we take the CoV of the distribution of post-ReLU activations. Then, for each network, we take the distribution of CoVs of its subclusters, and find the first quartile, median, and third quartile. For each training condition, we train five networks, rank the five by their median CoV, take the median network under this ranking, and report that network's CoV quartiles in the final three columns of table 6. We find that in some cases the CoVs are relatively low, including the ImageNet models which indicates relatively consistent activations. In other networks though, many of the CoVs are above 1.

| | Activational Variance | | Activational Coefs of Variation | | |
|---|---|---|---|---|---|
| **Network** | **Fisher Combined $p$** | **Chi Squared $p$** | **Quartile 1** | **Median** | **Quartile 3** |
| MLP MNIST | 0.124 | 0.295 | 1.399 | 1.539 | 1.877 |
| CNN MNIST | 0.485 | 0.328 | 0.614 | 0.691 | 0.787 |
| VGG CIFAR (Unreg) | 0.061 | $\mathbf{4.39 \times 10^{-4}}$ | 0.929 | 1.214 | 3.690 |
| VGG CIFAR | **0.023** | **0.003** | 0.866 | 1.153 | 1.384 |
| VGG-16 ImageNet | 0.091 | 0.327 | 0.424 | 0.528 | 0.748 |
| VGG-19 ImageNet | $\mathbf{7.44 \times 10^{-4}}$ | $\mathbf{1.13 \times 10^{-4}}$ | 0.367 | 0.453 | 0.561 |
| ResNet50 ImageNet | **0.016** | **0.002** | 0.650 | 0.753 | 0.810 |

Table 6: **Combined $p$ values testing for differences in variance and coefficient of variation quartiles in feature visualization experiments:** Compare to table 1a. In the activational variance columns, $p$ values are shown comparing true sub-cluster activations to random sub-cluster activations. As in table 1a, $p$ values for non-ImageNet networks are medians among 5 trials, and $p$ values less than 0.05 are bolded. In the coefficients of variance columns, quartiles for the CoVs for true sub-clusters are shown.

## A.9 LESION TESTS

Section 3.2 presents the lesion test experiments. Example accuracy-change profiles for an MLP and small CNN in the Fashion datasets are shown here in figure 6. Table 7 and Table 8 show data on the importance of sub-clusters in the single lesion experiments, and is plotted in figure 2. "Acc. diff." means the difference in accuracy between the actual network and the network with that sub-cluster lesioned, while "Acc. diff. dist." shows the mean and standard deviation of the distribution of accuracy differentials between the actual network and one with a random set of neurons lesioned.

The "Proportion" column denotes the proportion of the layer's neurons that the sub-cluster represents. 'Important' means that the sub-cluster makes up at least 5% of the layer, that the drop in accuracy is greater than one percentage point, and that it was more important than all of 20 random sub-clusters it was compared against. 'Sig-but-not-diff' means that the drop in accuracy is significant but less than 1 percentage point, 'Diff-but-not-sig' means that the lesioning damage was more than 1 percentage point but not significant, 'Prunable' means that the drop in the accuracy is smaller than all random shuffles and smaller than 1 percentage point, 'Complete' means that the sub-cluster contains the whole layer, 'Small' means that the sub-cluster consists of less than 5% of the layer, and 'Other' means that the drop in accuracy is not statistically significant and less than 1 percentage point.

One detail not included in the main paper is that for the sake of computational efficiency, two measures were used for lesion experiments in the ImageNet models we used (ResNet-18 and VGG-16). First, we used a downsampled version of the ImageNet2012 dataset (Krizhevsky et al., 2012) with 10,000 instead of 50,000 images. Second, we omitted sub-clusters with fewer than 5 neurons or more than 90% of the neurons in the layer (this is different from the thresholds of 3 units and 80% we used for input construction experiments).

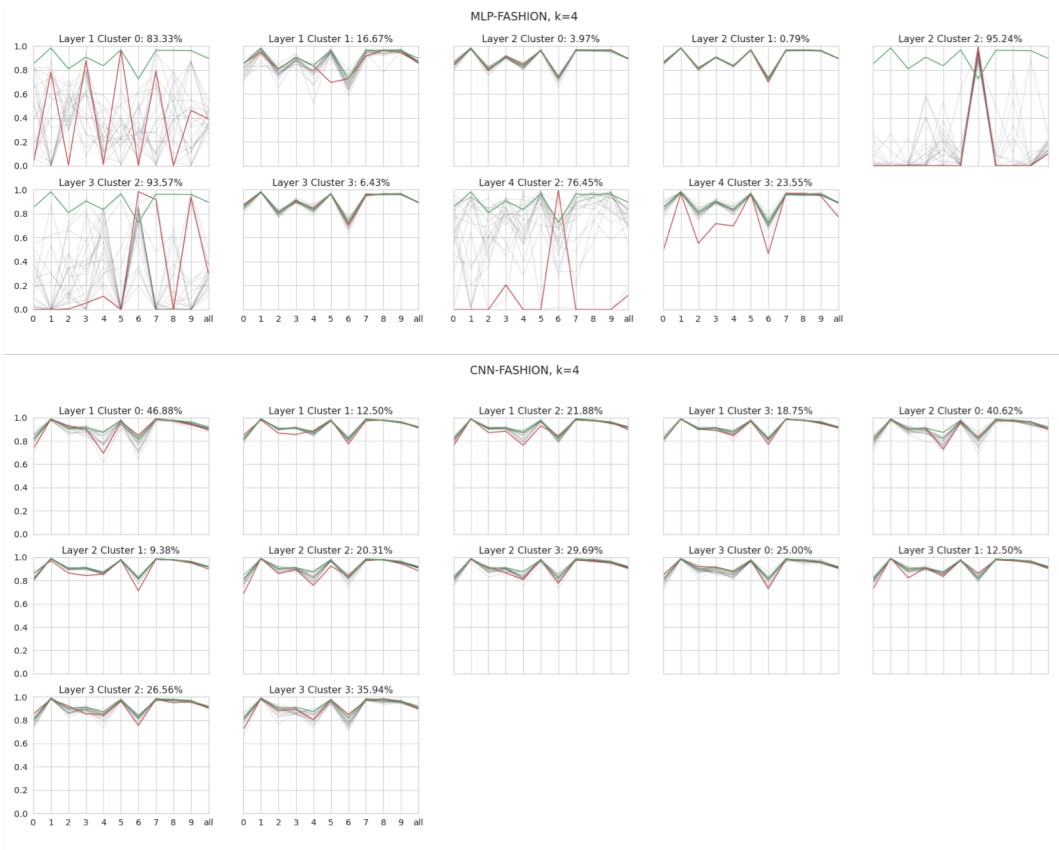

Figure 6: **Example accuracy change profiles for lesion tests:** (Top) Fashion-MNIST MLP and (Bottom) Fashion-MNIST Small CNN. Each subplot corresponds to a sub-cluster in the network. For the sake of visualization here, 4 clusters were used instead of the normal 12 which was used in all quantitative experiments. The accuracies for each class plus the overall accuracies in the rightmost points are plotted. Green lines indicate the performance of the unperturbed network, red lines give results when lesioning true sub-clusters, and grey lines show results for lesioning 20 random sub-clusters.

### A.10 EXPLORING THE "COMPOSABILITY" PROXY WITH DOUBLE LESION TEST

Given the lesion test presented in the main text, we know which sub-clusters are important, and it would be ideal to understand how the important sub-clusters depend on each other. To do this, we conduct experiments where we lesion two different important sub-clusters, which we'll call $X$ and $Y$, in different layers. First, we measure the loss in accuracy when both are lesioned, which we'll call $\ell(X \cup Y)$. We then compare $\ell(X \cup Y)$ to the loss in accuracy $\ell(X \cup Y')$ if we take a random subset $Y'$ of neurons of size $|Y|$ from the same layer as $Y$, and check if $\ell(X \cup Y)$ is larger than 50 random samples of $\ell(X \cup Y')$. This tests if the damage from lesioning $Y$ is statistically significant given how many neurons are contained in $Y$, and given that we are already lesioning $X$. We also calculate $\delta(Y, X) := \ell(X \cup Y) - \ell(X)$, which is the additional damage from lesioning $Y$ given that $X$ has been lesioned. If $\ell(X \cup Y)$ is statistically significantly different to the distribution of $\ell(X \cup Y')$, and if $\delta(Y, X)$ is larger than one percentage point, we say that sub-cluster $Y$ is important

Table 7: **Table of sub-clusters of an MLP network trained on Fashion-MNIST using pruning and dropout.** The network is partitioned into 8 clusters.

| Layer | Label | Acc. diff. | $p$ value | Proportion | Type | Acc. diff. dist. |
|---|---|---|---|---|---|---|
| 1 | 0 | 0 | 0.476 | 0.005 | small | $-0.00009 \pm 0.00021$ |
| 1 | 1 | $-0.2484$ | 0.048 | 0.404 | important | $-0.00751 \pm 0.0048$ |
| 1 | 2 | $-0.0685$ | 0.048 | 0.154 | important | $-0.00179 \pm 0.00222$ |
| 1 | 3 | $-0.0016$ | 0.429 | 0.111 | other | $-0.00124 \pm 0.00105$ |
| 1 | 5 | 0 | 0.429 | 0.005 | small | $-0.00011 \pm 0.00035$ |
| 1 | 6 | $-0.1761$ | 0.048 | 0.313 | important | $-0.00453 \pm 0.00315$ |
| 1 | 7 | 0 | 0.476 | 0.01 | small | $-0.00006 \pm 0.00039$ |
| 2 | 0 | $-0.0134$ | 0.048 | 0.473 | important | $-0.00362 \pm 0.00296$ |
| 2 | 1 | 0 | 0.476 | 0.008 | small | $0.00002 \pm 0.00025$ |
| 2 | 2 | $-0.0023$ | 0.048 | 0.027 | small | $0.000005 \pm 0.00038$ |
| 2 | 3 | $-0.006$ | 0.048 | 0.113 | sig-but-not-diff | $-0.00045 \pm 0.00114$ |
| 2 | 6 | $-0.0014$ | 0.476 | 0.188 | other | $-0.00126 \pm 0.00147$ |
| 2 | 7 | 0.0014 | 1 | 0.191 | prunable | $-0.00143 \pm 0.00131$ |
| 3 | 0 | $-0.0343$ | 0.048 | 0.617 | important | $0.0005 \pm 0.00331$ |
| 3 | 4 | 0.0005 | 0.905 | 0.023 | small | $-0.00002 \pm 0.0003$ |
| 3 | 7 | $-0.0129$ | 0.048 | 0.359 | important | $0.00202 \pm 0.00186$ |
| 4 | 0 | $-0.0379$ | 0.143 | 0.563 | diff-but-not-sig | $-0.02009 \pm 0.02002$ |
| 4 | 3 | 0.0002 | 1 | 0.004 | small | $-0.00005 \pm 0.00017$ |
| 4 | 4 | $-0.0017$ | 0.095 | 0.113 | other | $-0.00089 \pm 0.00086$ |
| 4 | 7 | $-0.2048$ | 0.048 | 0.32 | important | $-0.00445 \pm 0.00138$ |

Table 8: **Table of sub-clusters of an VGG network trained on CIFAR-10 using dropout.** The network is partitioned into 8 clusters.

| Layer | Label | Acc. diff. | $p$ value | Proportion | Type | Acc. diff. dist. |
|---|---|---|---|---|---|---|
| 1 | 7 | $-0.8043$ | 0.048 | 1 | complete | $-0.80430$ |
| 2 | 5 | 0 | 0.857 | 0.016 | small | $-0.00532 \pm 0.00708$ |
| 2 | 7 | $-0.8043$ | 0.143 | 0.984 | diff-but-not-sig | $-0.80470 \pm 0.00166$ |
| 3 | 1 | $-0.8043$ | 0.048 | 1 | complete | $-0.80430$ |
| 4 | 1 | $-0.8043$ | 0.048 | 1 | complete | $-0.80430$ |
| 5 | 1 | $-0.8043$ | 0.048 | 1 | complete | $-0.80430$ |
| 6 | 1 | $-0.8043$ | 0.048 | 1 | complete | $-0.80430$ |
| 7 | 1 | $-0.8043$ | 0.048 | 0.617 | important | $-0.12236 \pm 0.03685$ |
| 7 | 5 | 0 | 1 | 0.383 | prunable | $-0.03632 \pm 0.01111$ |
| 8 | 0 | 0 | 1 | 0.586 | prunable | $-0.02803 \pm 0.01497$ |
| 8 | 1 | 0 | 0.238 | 0.002 | small | $-0.00005 \pm 0.00020$ |
| 8 | 5 | $-0.8043$ | 0.048 | 0.412 | important | $-0.00752 \pm 0.00451$ |
| 9 | 3 | 0 | 1 | 0.822 | prunable | $-0.21835 \pm 0.12462$ |
| 9 | 5 | $-0.8043$ | 0.048 | 0.178 | important | $-0.00200 \pm 0.00146$ |
| 10 | 2 | 0 | 1 | 0.924 | prunable | $-0.62427 \pm 0.15113$ |
| 10 | 6 | $-0.8043$ | 0.048 | 0.076 | important | $-0.00143 \pm 0.00289$ |
| 11 | 2 | 0 | 0.143 | 0.002 | small | $-0.00001 \pm 0.00004$ |
| 11 | 4 | 0 | 1 | 0.662 | prunable | $-0.05353 \pm 0.03084$ |
| 11 | 6 | $-0.8043$ | 0.048 | 0.336 | important | $-0.00387 \pm 0.00240$ |
| 12 | 4 | $-0.8043$ | 0.048 | 1 | complete | $-0.80430$ |
| 13 | 4 | -0.8043 | 0.048 | 1 | complete | $-0.80430$ |

conditioned on sub-cluster $X$. Similarly, we test if $X$ is important conditioned on $Y$ by comparing $\ell(X \cup Y)$ to the distribution of $\ell(X' \cup Y)$, and by determining the size of $\delta(X, Y)$. Table 9 shows the $\delta$ values and importances of different pairs of sub-clusters for an MLP trained on Fashion-MNIST with pruning and dropout, when the number of cluster is set to 8 for visualization.

By examining the importances of sub-clusters conditioned on each other, we can attempt to construct a dependency graph of sub-clusters by determining which sub-clusters send information to which

Figure 7: **Four cases of dependency between sub-clusters.** These cases characterize how information can flow from a sub-cluster $X$ to a sub-cluster $Y$, where $X$ is in an earlier layer than $Y$. $U$ and $V$ are two other sub-clusters, which reside in the same layer as $X$ and $Y$ respectively. All of these sub-clusters are individually "important". Given the importance configuration of $(X|Y, Y|X)$, we conjecture what is the relationship in terms of information flow between the sub-clusters.

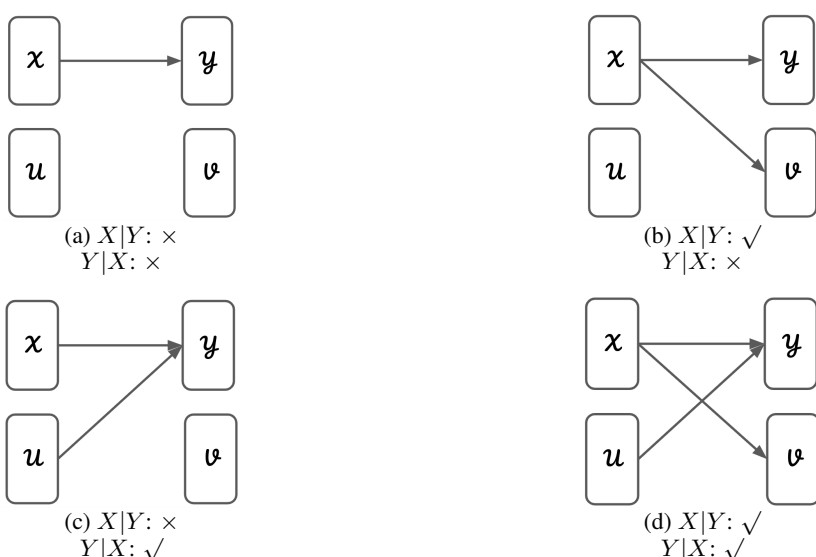

others. Consider a pair of sub-clusters $(X, Y)$ where $X$ is in an earlier layer than $Y$, and where both are individually important (refer to figure 7 for an elaborated visual illustration).

- If $X$ is not important conditioned on $Y$, and $Y$ is not important conditioned on $X$, we reason that all of the information from $X$ is sent to $Y$ (since otherwise lesioning $X$ would damage accuracy even conditioned on $Y$ being lesioned), and that the only information that $Y$ receives is sent via $X$ (since otherwise lesioning $Y$ would damage accuracy even conditioned on $X$ being lesioned).

- If $X$ is not important conditioned on $Y$ but $Y$ is important conditioned on $X$, then we reason that $X$ sends information to $Y$ and also to other sub-clusters.

- If $Y$ is not important conditioned on $X$ but $X$ is important conditioned on $Y$, we reason that $Y$ receives information from $X$ and other sub-clusters.

- We can draw no conclusion if both $X$ and $Y$ are important conditioned on the other.

These assumptions, together with data shown in figure 9, let us draw some edges in a dependency graph of sub-clusters, which is shown in figure 8. Note that sub-clusters of cluster 0 seem to send information to each other, which is what we would expect if modules were internally connected. The same holds for the sub-cluster of cluster 7.

Table 9: **Dependency information of pairs of "important" sub-clusters of an MLP trained on Fashion-MNIST with dropout.** $X$ and $Y$ represent sub-clusters, $X$ being the one earlier in the network. They are numbered by first their layer and then their cluster number. The network is partitioned into 8 cluster.

| $\mathbf{X}$ | $\mathbf{Y}$ | $\mathbf{X|Y}$ | $\mathbf{Y|X}$ | $\delta(\mathbf{X}, \mathbf{Y})$ | $\delta(\mathbf{Y}, \mathbf{X})$ |
|---|---|---|---|---|---|
| 1-1 | 2-0 | $\checkmark$ | $\checkmark$ | -0.261 | -0.026 |
| 1-1 | 3-0 | $\checkmark$ | $\checkmark$ | -0.237 | -0.023 |
| 1-1 | 3-7 | $\checkmark$ | $\times$ | -0.249 | -0.014 |
| 1-1 | 4-7 | $\checkmark$ | $\checkmark$ | -0.244 | -0.201 |
| 1-2 | 2-0 | $\checkmark$ | $\checkmark$ | -0.095 | -0.04 |
| 1-2 | 3-0 | $\checkmark$ | $\checkmark$ | -0.092 | -0.058 |
| 1-2 | 3-7 | $\checkmark$ | $\times$ | -0.054 | +0.001 |
| 1-2 | 4-7 | $\checkmark$ | $\checkmark$ | -0.088 | -0.224 |
| 1-6 | 2-0 | $\checkmark$ | $\times$ | -0.157 | 0.006 |
| 1-6 | 3-0 | $\checkmark$ | $\times$ | -0.194 | -0.052 |
| 1-6 | 3-7 | $\checkmark$ | $\checkmark$ | -0.180 | -0.017 |
| 1-6 | 4-7 | $\checkmark$ | $\checkmark$ | -0.153 | -0.182 |
| 2-0 | 3-0 | $\times$ | $\times$ | -0.125 | -0.146 |
| 2-0 | 3-7 | $\checkmark$ | $\checkmark$ | -0.125 | -0.124 |
| 2-0 | 4-7 | $\times$ | $\checkmark$ | -0.101 | -0.292 |
| 3-0 | 4-7 | $\checkmark$ | $\checkmark$ | -0.502 | -0.673 |
| 3-7 | 4-7 | $\times$ | $\checkmark$ | -0.010 | -0.202 |

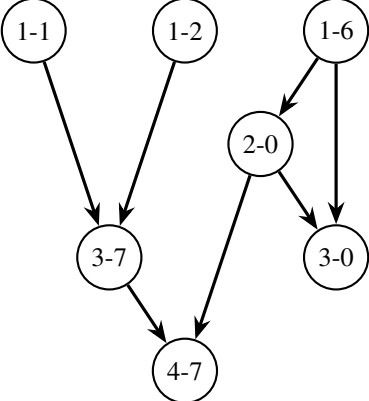

Figure 8: **A plot of the edges of the sub-cluster dependency graph for an MLP trained on Fashion-MNIST and partitioned with 8 clusters.** We derive this from data shown in figure 9. Sub-clusters are identified first by their layer number, then by their module number.

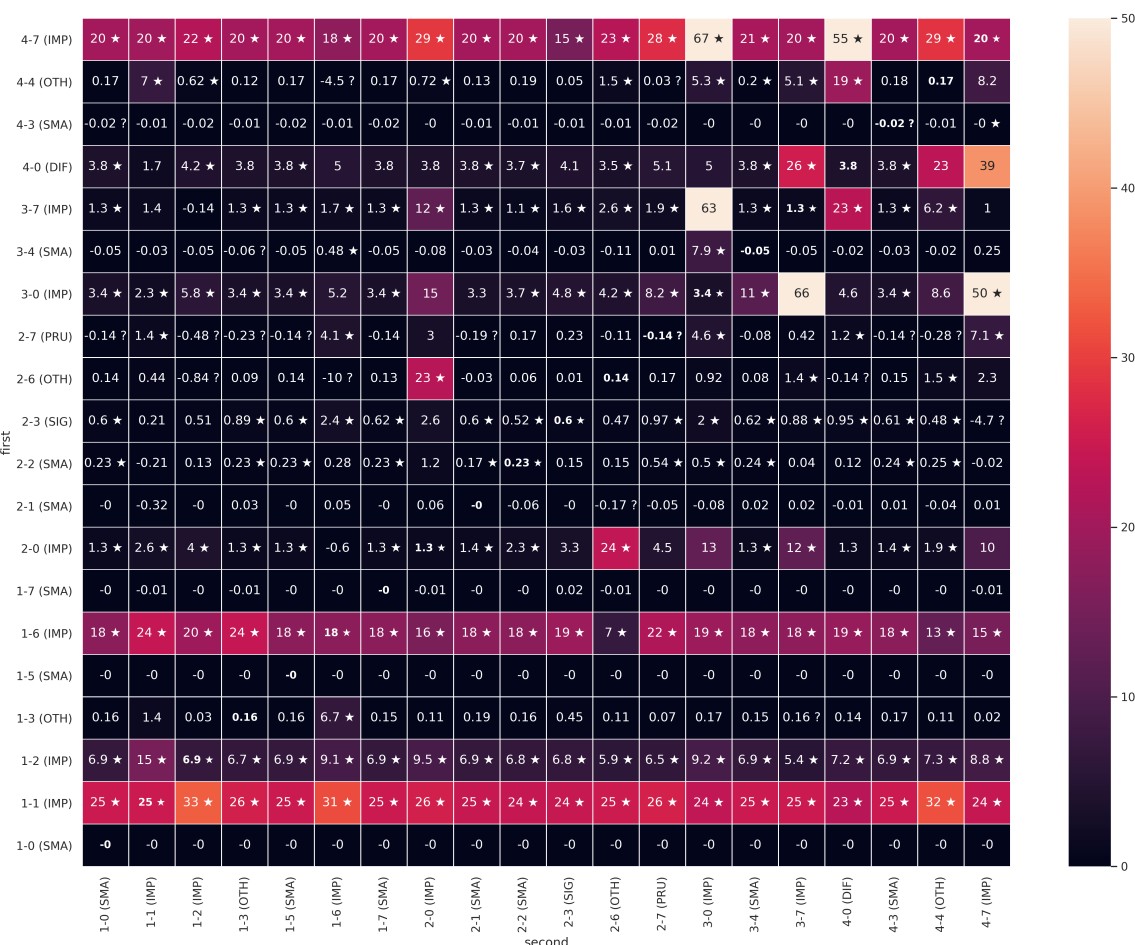

Figure 9: **Dependency information for all pairs of sub-clusters for an MLP trained on Fashion-MNIST and partitioned with 8 clusters.** Best viewed in color and zoomed in on a screen. Sub-clusters are numbered by their layer and cluster. They are also labeled by their importance: IMP stands for "important", SMA stands for "small", SIG stands for "sig-but-not-diff", DIFF stands for "diff-but-not-sig", PRU stans for "prunable" and OTH stands for "other", terms which are defined at the beginning of Appendix A.9. Cells on the diagonal are labeled by the accuracy damage in percents caused by zeroing out the corresponding sub-cluster in the single lesion experiments, and have bolded text, while cells off the diagonal are labeled by $100 \times \delta(\text{first}, \text{second})$. Cells contain stars if the damage caused by lesioning the first is statistically significant at the $p < 0.05$ level conditional on the second being lesioned, and question marks if the damage caused by lesioning the first was less than the damage caused by lesioning each of 50 random sets of neurons (i.e., $p = 1.00$), conditional on the second being lesioned. Note that this plot includes sub-clusters in the same layer, where the meaning is questionable since there can be no direct dependency.

