# OpenReview forum: "Importance and Coherence: Methods for Evaluating Modularity in Neural Networks"
_ICLR.cc/2021/Conference — Reject_

### Official Review · AnonReviewer1 · 2020-10-27
**Interesting work,  some aspects need to be polished**

**Rating:** 5
**Confidence:** 4

**Review:**

The manuscript introduces an approach, based on importance and coherence, for evaluation whether a partitioning of a network exhibits modular characteristics.
Importance refers to how crucial is a neuron , or set of neurons, to the performance of a network on a given task, e.g. classification.
Coherence refers to how consistently the neuron(s) in question are related to specific features.
Experiments are conducted by considering sets of neurons identified via a spectral clustering algorithm.

The manuscript proposes a method to verify to what extent a partitioning of a network follows modular characteristics. To a good extent the proposed method is grounded on proper theoretical foundations which is highly desirable. The only part where this cannot be fully verified is its dependence on the method from [Anonymous, 2021] which cannot be verified.

My main concerns with the manuscripts are the following:

- When conducting spectral clustering, the number of clusters is set to 12. Is there a procedure to set this value in a principled manner? is there an indication on the effect of this parameter? the manuscript would benefit from analyzing the effect of this parameter in the observation made on the reported experiments?

- Visualizations discussed in Sec. 3.1.1 (Fig.1) are quite subjective. While in some cases some patterns are indeed visible, in other cases it is hard to make sense of what is being presented. Is it possible to evaluate the produced visualizations in a more objective manner?
In recent years, several methods ( Bau et al, 2016, Oramas et al. ICLR'19, Yang and Kim, arXiv:1907.09701 ) for quantitative evaluation of methods for visual interpretation and explanation have been processed. Perhaps one of these could be adopted in the manuscript with the goal of objectively evaluating the visualizations/explanations presented in Fig. 1.

- In some cases design decision are made that seem to favor observations expected in some experiments. For instance, in order to favor clusterability small MLPs are pruned, to improve visualization MLPs are trained with dropout; and other factors relevant to the proposed method. Therefore my question by ensuring that some of these properties, e.g. cleaner cluster, clearer visualizations, don't you favor the measurement capabilities of the proposed importance/coherence metrics?

- When analyzing the "importance" metric on the lesion tests (Sec. 3.2) there are new conditions that are applied to the clusters being considered in the analysis, e.g. the size of the cluster, minimum effect of the cluster on accuracy, etc. Keeping this present, my questions are: i) Were these conditions also applied when analyzing "coherence", and ii) why these type of condition were not applied in the experiments of Sec. 3.1? Ideally, some level of consistency is expected among the experiments. Otherwise it is hard to assess properly the origin of observations made on the results of the experiments.

- At the end of Sec. 4, it is stated that the conducted experiments, combining spectral clustering with feature visualization,
highlight the usefulness of combining multiple interpretability methods in order to build an improved set of tools for rigorously understanding systems. However, from the observations made on the experiments I do not see the added value that the proposed method could bring to interpretability/explainability of the analyzed models (networks).

- Very related, one paragraph later, it is stated that having modular networks is useful both for interpretability and for building better models. However, from the content of the manuscript it is not clear how having a modular network/representation does contribute with the two listed aspects.

- Significant parts of the manuscript are delegated to the supplementary material. In addition, the third part of the proposed method, i.e. intrinsic partition evaluation, is part of another manuscript [Anonymous,2021] that does not seem to be published. For these reasons, to a good extent, the manuscript is not self-contained.

---

> ### Author Response · Authors · 2020-11-14
> **Reply to reviewer 1**
>
> We thank reviewer 1 for the constructive feedback.
>
> (1) Re: “The only part where this cannot be fully verified is its dependence on the method from [Anonymous, 2021] which cannot be verified... the third part of the proposed method, i.e. intrinsic partition evaluation, is part of another manuscript [Anonymous,2021] that does not seem to be published”
> While the other paper investigates spectral clustering in depth, this paper uses spectral clustering as an example of a partition-generating procedure. In that sense, this paper can stand alone. Regarding intrinsic partition evaluation, we mention this for completeness, but it is not relevant to the experiments we conduct in this paper. However, we are hoping to improve clarity here. Do you have any improvements in mind? For example, is it confusing to mention Intrinsic Partition Evaluation in section 2?
>
> (2) Re: “When conducting spectral clustering, the number of clusters is set to 12.”
> We will change section 2 to clarify this. We wanted a number greater than 10 (the number of classes in MNIST/CIFAR), but small relative to the number of filters in the networks' layers. Note that finding results with k=12 in networks ranging from the MNIST to ImageNet scale suggests that results are robust to different choices of $k$. We have explored using higher $k$, but much larger values lead to slow runtimes for ImageNet experiments. We are currently running lesion experiments for k=8 and k=18 which we will report here soon.
>
> (3) Re: “Visualizations discussed in Sec. 3.1.1 (Fig.1) are quite subjective...Is it possible to evaluate the produced visualizations in a more objective manner?”
> We intended section 3.1.1. to provide simple visual examples before the more rigorous quantitative results in the rest of section 3. Note that in 3.1.2 and 3.2, we also perform experiments on the same types of networks which are visualized in 3.1.1.
>
> (4) Re: “In some cases design decision are made that seem to favor observations expected in some experiments...don't you favor the measurement capabilities of the proposed importance/coherence metrics”
> Would you be able to further clarify if/how this is an issue? We do not see an inherent issue with tuning our training process for clearer results. We will clarify in section 2 our approach with pruning and dropout. Put more simply, we use pruning in MLPs and no other networks. We use dropout in the regularized CIFAR-10 VGGs and also in the MLPs used for correlation-based visualization (with the exception of the network trained on halves-diff data in figure 4d). Also bear in mind that the baselines we compare all results to are random sub-clusters, not subclusters of unregularized networks.
>
> (5) Re: “When analyzing the "importance" metric on the lesion tests (Sec. 3.2) there are new conditions that are applied to the clusters being considered in the analysis...i) Were these conditions also applied when analyzing "coherence", and ii) why these type of condition were not applied in the experiments of Sec. 3.1?”
> We will rewrite some of these explanations for clarity. In all experiments with importance and coherence from which Table 1 is produced, we omit sub-clusters that are extremely small or large (details in supplement), but we do not otherwise select for clusters. We ONLY introduce new conditions for classifying sub-clusters for the sake of producing Figure 2 and providing an example taxonomy that demonstrates the diversity among sub-clusters in size, importance, and importance relative to random sub-clusters.
>
> (6) Re: “it is not clear how having a modular network/representation does contribute with the two listed aspects”
> Our argument is that this work motivates building networks that better lend themselves to modular deconstructions because this level of abstraction is useful for interpretability. The argument that modularity leads to better models made via discussion and citation of works in the final paragraph of related works.
>
> (7) Re: “I do not see the added value that the proposed method could bring to interpretability/explainability of the analyzed models.”
> Our argument is that the advantage of combining clustering with other interpretability methods is that it provides a method of generating partitions based on weight connectivity which doesn't involve activations or gradients. The key thing that weight-based clustering brings to the table is non-redundancy with methods based on runtime analysis.

---

> > ### Comment · AnonReviewer1 · 2020-11-22
> > **RE: Reply to reviewer 1**
> >
> > Thanks for the provided feedback to my initial reviews.
> >
> > It had clarified some of my initial doubts/questions
> >
> > Regarding point (1),  you mentioned that this paper uses spectral clustering as an example of a partition-generating procedure. So, are the observations made on the experiments purely exclusive to this method? if so, the manuscript would benefit from stating how this partitioning differs when using the followed method in comparison with other concept-based methods for model explanation.
> > It is not confusing to mention Intrinsic Partition Evaluation in Sec. 2. However, since it is presenting without major context (the related work section is at the end of the paper), the reader may get the impression that this paper is proposed a an interpretable partioned method based on spectral clustering. This second part, spectral clustering, being covered in depth by the other manuscript.
> >
> > I stressed point (4) from the perspective that aiming at evaluating factors A, B, C in different experiments and finetunning the method for each experiment optimizing each of these factors (separately) would definitively favor the method being optimized by providing clearer results for that method.
> > I got the impression this was the case from Sec. 3.1.1 (Fig. 1) where the different sub-clusters are assessed qualitatively, by visualization. Here dropout is used to improve visualization of the proposed method.
> >
> > Here, it would be desirable to have a fixed setting for the proposed method (and other baselines/compiting methods) and evaluate its different aspects with the different experiments followed in the paper. Perhaps this is indeed the procedure followed in the paper (and not the one mentioned above), if so, I would suggest to make very clear the fact that the methods considered in all the experiments reported in Sec.3.
> >
> > Regarding point (6), on the one hand, I agree that a partition-generating procedure or an algorithm for learning disentangled representations could be useful for enabling model interpretation and explanation. Existing works in the literature have proved that already (e.g. Ghorbani et al., NeurIPS'19). On the other hand, I find weak the connection of the  partition-generating procedure proposed in the manuscript and model interpretation. The lesion tests (Sec.3.2) suggest that the identified partitions (sub-clusters) are relevant for the predictions of the model. However, the visualizations in Sec.3.1 look quite subjective to me. Therefore my previous request for an objective comparison of these visualizations.

---

> > > ### Author Response · Authors · 2020-11-25
> > > **Response to Reviewer 1**
> > >
> > > We thank Reviewer 1 for the additional comments.
> > >
> > > (1) +(6)  Indeed, we use only spectral clustering as a partition-generation method. Although our approach may be related to disentangled representations, we are using standard training methods and partitioning the network purely based on the weight values, and are unaware of prior research suggesting that the resulting partitioning of the network will have properties that would be useful for model interpretation and explanation. Similarly, while Ghorbani et al find visual concepts that networks depend on, they do not locate regions in the network that respond to these concepts, nor do they propose ways of evaluating regions for modularity or find any interesting results from this evaluation. Their criteria of importance and coherence is a property of a visual concept (input) and not of groups of neurons.
> > >
> > > (4) In updates to the paper, we rewrote the fourth paragraph of section 3 to clarify our approach to pruning and dropout. In summary, pruning was used for MLPs, and dropout was used to produce correlation-based visualization examples. The only other time in which we used dropout was for the regularized CIFAR-10 VGGs whose results are presented alongside the unregularized ones.

---

### Official Review · AnonReviewer4 · 2020-10-27
**Interesting proposal to inspect the emergent structure in deep networks, which should be refined and compared with alternative methods based on graph-theory**

**Rating:** 4
**Confidence:** 4

**Review:**

This paper explores the application of spectral clustering methods to assess modular organization in the emergent architecture of deep networks. In particular, sub-modules identified by spectral clustering are evaluated in terms of "importance" and "coherence", two metrics defined by the authors with the goal of capturing how crucial the neurons in the sub-module are to the classification accuracy, and how consistent their activation is across input and output patterns.
Overall, the paper addresses important questions related to the way structural properties in a deep network might support the emergence of functional properties, which is a key issue given the relatively poor theoretical understanding we have about these self-organizing systems. The paper is comprehensible, though the general structure and the writing could be improved to improve readability. For example, in Section 3 it is not immediately clear how the importance and coherence metrics relate to the specific technique adopted for feature visualization, or to the lesioning method applied. The “Related Work” section should be moved at the beginning of the paper, and the contribution should be better framed in the context of other existing approaches.
Although I appreciate the wide range of networks tested by the authors, I think that the way results are presented does not easily allow to establish whether we can identify a series of robust findings that are valid across all architectures / tasks, or whether specific cases entail peculiar findings. It would also be valuable to better assess the relationship between modularity and regularization techniques, such as dropout, L2 and pruning: I think that this is a very important point that should deserve further investigation, since it could give important insights about the role of regularizers in shaping the final network architecture.
Finally, it would strengthen the paper if the proposed method is put in relation to other approaches that have proven effective to address this kind of question. This would greatly improve the robustness of the results, since the current baseline is basically constituted by a comparison with random sub-modules. For example, it would be nice to see if clustering coefficient and average path length (as defined in [1]) can provide useful information also for the analyses proposed by the authors. Note that in [1] the authors investigated deep networks with similar architectures (MLPs, CNNs, ResNets) trained on similar tasks (CIFAR-10, ImageNet).

Other comments:
- Pg. 3: the technique used to visualize a sub-cluster by creating an aggregate measure of the learned features can be discussed in relation to the method based on Earth-mover distance proposed by [2], where the authors also discuss other graph-based metrics that might be useful in the present setting.
- Could the “intersection information” approach presented in [3] can be exploited also in the analyses of the sub-modules detected by the spectral clustering? Note that in [3] the authors also investigate “lesion tests” by means of interventional techniques, which would make that approach very interesting as a further benchmark.
- What is the rationale for setting the number of clusters to 12? If this value is not theoretically motivated, further analyses should show that the results are robust to variations in this value.
- When comparing “true sub-clusters” with “random sub-clusters”, a useful control analysis would be to create random sub-clusters by matching some connectivity property (e.g., same average strength, clustering coefficient and/or average path length).
- Regarding the gradient-based method discussed in Olah et al. (2017), it would be useful to have some dispersion measure over the final optimization score, in order to better assess whether all neurons in the sub-cluster where in fact similarly activated by the optimized image.
- It would be interesting to include as baseline some analysis on randomly connected networks, since it has been shown that subgraphs in large random networks can in fact support accurate task performance even without ever training the weight values [4].
- Where the p-values corrected for multiple comparisons?
- The authors consider “modularity  as an organizing principle to achieve mechanistic transparency”. Though I sympathize with this statement, I guess there are several cases where modular systems (or in general systems with localized representations) can develop complex emergent dynamics that still prevent interpretability.

References
[1]	J. You, J. Leskovec, K. He, and S. Xie, “Graph Structure of Neural Networks,” in International Conference on Machine Learning, 2020.
[2]	A. Testolin, M. Piccolini, and S. Suweis, “Deep learning systems as complex networks,” J. Complex Networks, vol. 0000, no. 1, pp. 1–21, Jun. 2019.
[3]	S. Panzeri, C. D. Harvey, E. Piasini, P. E. Latham, and T. Fellin, “Cracking the Neural Code for Sensory Perception by Combining Statistics, Intervention, and Behavior,” Neuron, vol. 93, no. 3, pp. 491–507, 2017.
[4]	V. Ramanujan, M. Wortsman, A. Kembhavi, A. Farhadi, and M. Rastegari, “What’s Hidden in a Randomly Weighted Neural Network?,” In Proceedings of the IEEE/CVF Conference on Computer Vision and Pattern Recognition, pp. 11893-11902. 2020.

---

> ### Author Response · Authors · 2020-11-14
> **Reply to reviewer 4**
>
> We thank reviewer 4 for the constructive feedback.
>
> (1) Re: “the way results are presented does not easily allow to establish whether we can identify a series of robust findings that are valid across all architectures / tasks.”
> It is true that we find evidence of modular sub-clusters in some networks but not in others. However (a) we don’t believe that finding different trends across different networks should be disqualifying, and (b) while we show that spectral clustering often identifies modular sub-clusters, the the most central purpose of this paper is to develop ways to measure and use importance and coherence for modularity evaluation.
>
> (2) Re: “It would also be valuable to better assess the relationship between modularity and regularization techniques, such as dropout, L2 and pruning.”
> While we do not study this thoroughly, note that our comparisons between the regularized and unregularized CIFAR-10 VGGs relate to this. Nonetheless, we believe that further work focusing on this question and other ways to promote modularity architecturally or via training procedure will be valuable.
>
> (3) Re: “it would strengthen the paper if the proposed method is put in relation to other approaches that have proven effective to address this kind of question….For example, it would be nice to see if clustering coefficient and average path length...” and “When comparing “true sub-clusters” with “random sub-clusters”, a useful control analysis would be to create random sub-clusters by matching some connectivity property”
> Overall, we agree that additional controls would be interesting, but we do argue that controlling for location (layer) and size are sufficient to evaluate importance and coherence. In this paper, we do not aim to compare partition generation methods. As a side note, spectral clustering has a random-walk interpretation and is very closely related with commute-time/average path length. See von Luxburg (2007).
>
> (4) Re: Earth-mover distance and intersection information.
> We agree that earth mover distance could be used as a measure of coherence in 3.1.1. Other methods from 3.1.2 and 3.2 could as well. But we intended for 3.1.1. to give visual examples preceding more comprehensive results in 3.1.2 and 3.2. We also agree that intersection information could be used as a measure of coherence. However, to our understanding, because intersection information measures the association between a stimulus and output, this is closely related to our experiments with lesions. Is this consistent with your understanding?
>
> (5) Re: “What is the rationale for setting the number of clusters to 12?”
> We will change section 2 to clarify this. We wanted a number greater than 10 (the number of classes in MNIST/CIFAR), but small relative to the number of filters in the networks' layers. Note that finding results with k=12 in networks ranging from the MNIST to ImageNet scale suggests that these phenomena are robust to different choices of the ratio between $k$ and the size of the network. We have explored using higher $k$, but much larger values lead to slow runtimes for ImageNet experiments. We are currently running lesion experiments for k=8 and k=18 which we will report here soon.
>
> (6) Re: “Regarding the gradient-based method discussed in Olah et al. (2017), it would be useful to have some dispersion measure over the final optimization score”
> We agree this may be interesting. We are running some experiments to get a sense of dispersion and will hopefully be able to report this soon.
>
> (7) Re: “It would be interesting to include as baseline some analysis on randomly connected networks.”
> We agree that this would be interesting for understanding how much sub-cluster importance and coherence result from training, but we do not see a strong motivation for this involving the development of interpretability methods.
>
> (8) Re:  Where the p-values corrected for multiple comparisons?
> No. But we agree that also reporting multiple testing corrections in the Supplement would be helpful. We are working on this now and will report results soon.
>
> (9) Re: “I guess there are several cases where modular systems (or in general systems with localized representations) can develop complex emergent dynamics that still prevent interpretability.”
> We agree. We believe that future work will be needed for clarifying the association between interpretability and modularity. Note that this is a _motivation_ for our work though because developing methods for identifying modules and validating their boundaries is a prerequisite to answering this question.
>
> [1] Ulrike von Luxburg. A tutorial on spectral clustering. Statistics and computing, 17(4):395–416, 2007.

---

> > ### Comment · AnonReviewer4 · 2020-11-17
> > **Appropriate answers to my questions; I am looking forward to see the new results and further comparisons**
> >
> > (1) Re: Re: I agree with the authors that finding different trends across different networks should not be disqualifying; however, given that the focus of this method is to promote “interpretability”, it would be useful to at least give some possible explanations about why we might observe different outcomes in different architectures. What would be a possible strategy to promote the emergence of modularity in a network?
> >
> > (2) Re: Re: This is indeed related to the comment above; discovering that modularity emerges under some regularization pressure or learning constraints would make this approach much more valuable and useful in practice.
> >
> > (3) Re: Re: I acknowledge that implementing and testing all the possible alternative methods proposed by the Reviewers is unfeasible, and that should not be the aim of the review process. However, I also think that relatively simple graph-theory measures (such as clustering coefficient and average path length) or the consideration of control architectures (with random sub-clusters matched in connectivity properties) would allow to better understand how the proposed method compares with existing ones, and how robust it is under controlled variations in connectivity.
> >
> > (4) Re: Re: I think that the authors could just mention the techniques based on earth-mover distance (Testolin et al., 2019) and intersection information (Panzeri et al., 2017) as related work, given that these methods share many commonalities with the present approach. I agree that intersection information is more closely related to experiments with lesions, since it is based on interventions.
> >
> > (7) Re: Re: The authors argue that their focus is on the development of interpretability methods. However, I think that in order to demonstrate the effectiveness of an interpretability method it is also important to test it against alternative methods and baseline models, since these analyses can give further insights about the conditions where the method is most effective.

---

> > > ### Author Response · Authors · 2020-11-21
> > > **Reply to Reviewer 4**
> > >
> > > We thank Reviewer 4 for the additional comments and feedback.
> > >
> > > (1) We agree with the reviewer that this is a relevant question. Nevertheless, this paper focuses on developing the methods, and applying that one possible partition generation algorithm.
> > > We note that in the concurrent submission we analyzed the effect of various regularization and initialization on _structural_ modularity, i.e., without relation to the network performance as we did in this paper.
> > >
> > > (2) + (3) Re: We thank the reviewer for this suggestion. We think it is an interesting direction for future work.
> > >
> > > (4) Re: Thank you for pointing out these papers. We have added them to the related work.
> > >
> > > (7) Re: Currently we are running experiments on randomly-initialized VGGs. We’ll update when the new version of the paper is uploaded.

---

### Official Review · AnonReviewer3 · 2020-10-28
**Worthwhile ideas and concept, but feels incomplete**

**Rating:** 4
**Confidence:** 3

**Review:**

The authors identify putative clusters of units/neurons in deep networks using spectral clustering on a graph defined by synaptic weights. The authors then argue that these structurally defined clusters of neurons have similar *functional representations*. Finding interpretable relationships between weight matrices and functional modules is challenging, and the authors should be applauded for attempting to tackle this challenging problem that few research groups are devoting energy to.

Despite these positive notes, I have reservations about the presentation and results of the paper. My main concerns are:

(1) The results are largely qualitative and anecdotal. In figure 1, for example, the authors show slightly higher contrast in their identified clusters than random clusters. The results are limited to black and white images (MNIST and fashion-MNIST), and not all examples look great. Only 2 figures are shown in the main paper, with a lot of other details shoved into the supplement. Thus, the writing and presentation could be improved to highlight the most exciting and surprising findings.

(2) The results crucially rely on a second paper which was concurrently submitted and can't be reviewed because it is anonymized. The results shown in this paper are thin and qualitative (see point 1), so in my view these two paper should be combined into a single paper which overall might tell a more comprehensive and compelling story.

(3) The paper does not generate testable predictions or practical insights that could be used by used by practitioners. The only takeaway point for me was that some neurons / units show correlated representations, which is arguably already known (e.g. Csordas et al 2020). How to exploit this modularity to develop human-interpretable explanations of network function remains unclear to me.

---

> ### Author Response · Authors · 2020-11-14
> **Reply to reviewer 3**
>
> We thank reviewer 3 for the constructive feedback.
>
> (1) Re: “ Largely qualitative and anecdotal...The results shown in this paper are thin and qualitative”
> Section 3.1.1 and Figure 1 are indeed qualitative. Our goal for them was to provide visual examples with a simple dataset like MNIST. More importantly though, 3.1.2 and 3.2 are built entirely around quantitative results. We would like to emphasize that having quantitative results based on statistical hypothesis testing at all is actually something which sets this paper apart from related work using similar methods such as Bau et al. (2017), Watanabe (2019), and Carter et al. (2019).
>
> (2) Re: “The results [in Figure 1] are limited to black and white images (MNIST and fashion-MNIST), and not all examples look great.”
> What we want to emphasize in Figure 1is that these sub-clusters, despite being based only on weights, systematically exhibit coherence w.r.t. testing data. We find this interesting because it shows that we can uncover runtime properties of a network without making queries to it. And we believe this is useful because interpreting units at the sub-cluster level gives us a flexible level of abstraction and pairs well with data-sensitive interpretability methods.
>
> (3) Re: “Only 2 figures are shown in the main paper, with a lot of other details shoved into the supplement. Thus, the writing and presentation could be improved...”
> Would it be possible to provide more detailed suggestions about what figures and details from the supplement should be emphasized more and included in the main paper? For example, should we incorporate hypothesis testing details from the supplement into the main paper?
>
> (4) Re: “The results crucially rely on a second paper which was concurrently submitted and can't be reviewed because it is anonymized.”
> While the other paper investigates spectral clustering in depth, this paper only relies on spectral clustering as an example of a partition-generating procedure for neurons. In that sense, this paper stands alone. However, we are hoping to improve clarity here. What kind of details would you like to see? For example, is it confusing to mention Intrinsic Partition Evaluation in section 2?
>
> (5) Re: “The paper does not generate testable predictions or practical insights...”
> We agree that our main contribution is not a set of methods that will be immediately useful to practitioners, but this was not our goal. Our approach focused on understanding phenomena and introducing tools which evaluate whether a given neuron partitioning reflects the network functionality, and thus supports the existence of modularity.
>
> (6) Re: “The only takeaway point for me was that some neurons / units show correlated representations”
> This is not how we would characterize our contributions. While a cluster of units being highly correlated with one another w.r.t. a data distribution would be sufficient to show input coherence, it would not be necessary, nor would it establish importance or output coherence. Note that it is also novel that spectral clustering can find sets of neurons which exhibit these properties at all. We argue that the main takeaways from this paper are the concepts of importance and coherence, statistical methods for measuring them, and demonstrating that spectral clustering, which takes only the network’s weights as an input, often reveals important and input coherent sub-clusters.
>
> [1] David Bau, Bolei Zhou, Aditya Khosla, Aude Oliva, and Antonio Torralba. Network dissection: Quantifying interpretability of deep visual representations. In Proceedings of the IEEE conference on computer vision and pattern recognition, pp. 6541–6549, 2017.
> [2] Chihiro Watanabe. Interpreting layered neural networks via hierarchical modular representation. In International Conference on Neural Information Processing, pp. 376–388. Springer, 2019.
> [3] Shan Carter, Zan Armstrong, Ludwig Schubert, Ian Johnson, and Chris Olah. Activation atlas. Distill, 4(3):e15, 2019.

---

> > ### Comment · AnonReviewer3 · 2020-11-17
> > **Good Responses**
> >
> > I am struggling with what to assign for my final score because (a) I think the paper has a unique perspective and brings up some potentially interesting ideas that are understudied, but (b) the results are still thin (no CIFAR, ImageNet, etc.) and the meaning / significance of the clusters is not well established. It does not seem like each module had a very simple and interpretable functional role (see Figs 1 & 5), so the motivation of the paper in the abstract (transparency and safety) ring hollow to me. I also still think this should either be merged with the concurrent submission or make no mention of the other paper whatsoever.
> >
> > (Re: 1) I don't find the statistical hypothesis testing particularly exciting. It is a useful contribution, and helps me believe the clusters are real. But it doesn't establish that the results are a functionally useful way of thinking about understanding the inner workings of the network. Further, it is not clear to me that what you are picking up on are really clusters or heavy tails / spikes in a high-D distribution over activations. I think the details on the Fisher and Chi-Squared tests could be relegated to the Supplement to make room for more results.
> >
> > (Re: 2) I agree the result is interesting, but I have to squint to see it. If there was a huge beautiful effect in a more complex network and task (e.g. CIFAR) I would be more excited about this work. As it stands I am not confident the result indicates something very important to the network's function... It might just be a quirk.
> >
> > (Re: 3) I would like to see CIFAR in Fig1. I would like to see Fig 2 reproduced across multiple networks including CIFAR. Also, the colors in Fig 2 are very hard for me to parse and interpret. It would also be useful to visualize the shuffle controls (e.g. through histograms) that went into computing table 1. It would also be useful to compare spectral clustering with other clustering methods. These are just a couple ideas. It is hard for me to make more specific suggestions, because my gut feeling is just that there needs to be "more there" in the paper.
> >
> > (Re: 4) I think just not citing the other paper at all would address my concern.
> >
> > (Re: 5) I agree this is not a large problem. It just is a limitation.
> >
> > (Re: 6) I agree with your response.

---

> > > ### Author Response · Authors · 2020-11-21
> > > **Reply to Reviewer 3**
> > >
> > > We thank Reviewer 3 for the additional comments and feedback.
> > >
> > > (Re 1:) We are a bit confused regarding the sentence: “Further, it is not clear to me that what you are picking up on are really clusters or heavy tails / spikes in a high-D distribution over activations.”
> > > Our method does not cluster activations, but rather neurons based on the network's weight only. The statistical hypothesis testing procedure allows us to establish a link between the partition and the network performance. Indeed, it does not help us to understand the inner-working of a cluster, but it does show that these clusters, which were found without any use of data, are important and coherent (compared to the control partitions), and it suggests them as candidates for further interpretability investigation.
> > > Also, since the hypothesis tests were based on percentiles which could only be as low as 0.1, our tests have a type natural robustness to outliers in a heavy tail.
> > >
> > > (Re: 3) We are producing a similar plot as Figure 2 for VGG models trained on CIFAR-10 and Imagenet. We’ll update when the new version of the paper is uploaded. Figure 6 in the appendix (p. 19) shows what we call “accuracy profile”, plots of overall and per-class accuracies for the lesioning of true sub-cluster as well as multiple random sub-clusters. Although it is not shown as a histogram format, it does demonstrate the distribution of accuracies of the random shuffle control.

---

> > > > ### Comment · AnonReviewer3 · 2020-11-21
> > > > **Clarification**
> > > >
> > > > My apologies, I should have said "high-D distribution over weights" -- see, for example, https://arxiv.org/abs/1901.08276

---

> > > > > ### Author Response · Authors · 2020-11-25
> > > > > **Response to Reviewer 3**
> > > > >
> > > > > We thank Reviewer 3 for clarifying this point.
> > > > > Martin et al. discuss heavy tail distributions of _singular values_ as a way to model the distribution of weight metrics. We are not aware of an empirical work that shows that weight distributions of trained neural networks are heavy-tailed.

---

### Author Response · Authors · 2020-11-14
**Comment to all reviwers: key points**

We would like to thank the reviewers for their thoughtful comments and efforts towards improving our paper. We responded to each reviewer in a separate comment, in addition, in this comment we address some of the most important points from the reviewers. Nevertheless, please refer to the individual comment for a detailed reply.

Our main results are summarized in table 1 and discussed in sections 3.1.2 and 3.2. These results are built entirely around a quantitative approach based on statistical hypothesis testing, and it is what sets this paper apart from related work using similar methods such as Bau et al. (2017), Watanabe (2019), and Carter et al. (2019). Indeed, section 3.1.1 and Figure 1 are qualitative, and they are intended to provide visual examples with a simple dataset like MNIST in analyzing clusters.

The reasoning that guided us in choosing the number of clusters to be $k=12$ is that we wanted a number greater than 10 (the number of classes in MNIST/CIFAR), but small relative to the number of filters in the networks' layers. We are currently running lesion experiments for k=8 and k=18 which we will report here soon.

On top of that, we run new experiments that measure dispersion over the final optimization score for the input cohorision test, following the suggestion of Reviewer #4. We are also adding to the paper analysis of p values under a multiple testing framework. We will also report on these soon.

In this paper we cite a concurrent submission about partition-generating using spectral clustering. All the relevant details are explained also in this paper, and we consider it self-contained in this sense, as the proposed modularity evaluation method can be applied to any partitioning of a neural network. We would like to ask the reviewers whether there is any detail that is missing.

Finally, we believe that modularity could be one possible path, in addition to others, for achieving mechanistic transparency of neural networks. The proposed methods in this paper for evaluating a network partitioning, are one step in this direction. We find it exciting that networks display modularity that can be found via spectral clustering. We think that the most valuable implications of this work in the future will involve modularity evaluation in networks with more explicitly modular architectures and training procedures.

[1] David Bau, Bolei Zhou, Aditya Khosla, Aude Oliva, and Antonio Torralba. Network dissection: Quantifying interpretability of deep visual representations. In Proceedings of the IEEE conference on computer vision and pattern recognition, pp. 6541–6549, 2017.
[2] Chihiro Watanabe. Interpreting layered neural networks via hierarchical modular representation. In International Conference on Neural Information Processing, pp. 376–388. Springer, 2019.
[3] Shan Carter, Zan Armstrong, Ludwig Schubert, Ian Johnson, and Chris Olah. Activation atlas. Distill, 4(3):e15, 2019.

---

### Author Response · Authors · 2020-11-18
**Updates for new revision**

In response to feedback, we have posted an updated version of the paper. The key updates are as follows:
* Section 2, first paragraph: we clarified that intrinsic partition evaluation is not relevant to the methods used in this paper.
* Section 2, final paragraph: we added an explanation of why we chose k=12 clusters.
* Section 3 paragraph 4: we clarified the explanation of when pruning and dropout were used.
* Section 3.1.1 paragraph 1: we clarified that the purpose of this section is to provide visual examples and build intuition preceding the quantitative results in the following subsections.
* Section 3.2, Importance portion: we switched the order in which we present the taxonomy and the network wide hypothesis tests for importance so as not to suggest that we filtered for sub-clusters in the hypothesis tests based on the descriptive taxonomical criteria.
* Section 4, paragraph 2: we now mention earth mover distance and intersection information in related works.
* Appendix 5: Lesion test results for k=8 (50% fewer) and k=18 (50% more) clusters. They are highly similar to Table 1b.
* Appendix 6: multiple testing corrections of table 1 using both the Bonferrroni-Holm and Benjamini-Hochberg techniques. Both show that we find that most results survive the corrections for multiple comparisons.

We do not yet but will soon have updates on dispersion analysis (see reviewer 4’s comments) for input construction experiments. We will post another update soon.

---

### Author Response · Authors · 2020-11-25
**Updates for last revision**

We thank our reviewers for engaging in the review process in the last two weeks.
We have added updates concerning three experiments:

1. We added a plot for VGG trained on CIFAR into Figure 2, where the sub-clusters are visualized according to the results of the lesion test. In particular, we see that also for this network there is diversity in the sub-cluster according to our descriptive taxonomy.

2. We added a new appendix section, Appendix 5, focusing on untrained networks. We perform feature visualization experiments but not lesion-based experiments because an untrained network will only have accuracy at the random guess baseline with and without lesions. The results suggest that these trained networks are considerably more input coherent than untrained ones.

3. We added to what is now Appendix 8 analysis of the variance of activations for units in true and random sub-clusters when their visualizations are passed through the network. We also analyze the coefficients of variation for the activations in true sub-clusters. The results suggest that in some networks, the neurons in true sub-clusters tend to be activated by their visualizations with lower variance, and random sub-clusters tend to be activated by their visualizations with higher variance. This is a sign that in these networks, coherence tends to happen via positive rather than negative associations. Meanwhile, coefficients of variation are sometimes relatively low, including for ImageNet models, but are often over 1 for other models.

---

### Decision · Program_Chairs · 2021-01-07
**Final Decision**

**Decision:**

Reject

**Comment:**

The paper present an approach for defending for, and search for, 'modularity' in neural networks, as a step to better interpretations of their functional structure. This is an interesting, and highly original approach, as recognised by the reviewers. However,  there was also some discussion about what exactly can be learned from the derived clusters/modules, and if and how they will lead to a better understanding of neural networks, or provide concrete ways of improving them.  While the authors addressed some issues during the review process, and provided additional results, the consensus (of all three reviewers) was finally that the paper did not reach the quality standards required by ICLR. I share this view-- the paper provides a refreshing perspective, but I still am not convinced that I see a clear, compelling 'use case' for their approach.